# HGF and TSG-6 Released by Mesenchymal Stem Cells Attenuate Colon Radiation-Induced Fibrosis

**DOI:** 10.3390/ijms22041790

**Published:** 2021-02-11

**Authors:** Benoît Usunier, Clément Brossard, Bruno L’Homme, Christine Linard, Marc Benderitter, Fabien Milliat, Alain Chapel

**Affiliations:** Service de Recherche en Radiobiologie et en Médecine Régénérative (SERAMED), Laboratoire de Radiobiologie des Expositions Médicales (LRMED), Institut de Radioprotection et de Sûreté Nucléaire (IRSN), Fontenay-aux-Roses, F-92260 Paris, France; benoit.usunier@gmail.com (B.U.); clement.brossard@irsn.fr (C.B.); bruno.lhomme@irsn.fr (B.L.); christine.linard@irsn.fr (C.L.); marc.benderitter@irsn.fr (M.B.); fabien.milliat@irsn.fr (F.M.)

**Keywords:** mesenchymal stem cells, colorectal, cancer, fibrosis, radiotherapy, inflammation, fibrosis, HGF, TSG-6

## Abstract

Fibrosis is a leading cause of death in occidental states. The increasing number of patients with fibrosis requires innovative approaches. Despite the proven beneficial effects of mesenchymal stem cell (MSC) therapy on fibrosis, there is little evidence of their anti-fibrotic effects in colorectal fibrosis. The ability of MSCs to reduce radiation-induced colorectal fibrosis has been studied in vivo in Sprague–Dawley rats. After local radiation exposure, rats were injected with MSCs before an initiation of fibrosis. MSCs mediated a downregulation of fibrogenesis by a control of extra cellular matrix (ECM) turnover. For a better understanding of the mechanisms, we used an in vitro model of irradiated cocultured colorectal fibrosis in the presence of human MSCs. Pro-fibrotic cells in the colon are mainly intestinal fibroblasts and smooth muscle cells. Intestinal fibroblasts and smooth muscle cells were irradiated and cocultured in the presence of unirradiated MSCs. MSCs mediated a decrease in profibrotic gene expression and proteins secretion. Silencing hepatocyte growth factor (HGF) and tumor necrosis factor-stimulated gene 6 (TSG-6) in MSCs confirmed the complementary effects of these two genes. HGF and TSG-6 limited the progression of fibrosis by reducing activation of the smooth muscle cells and myofibroblast. To settle in vivo the contribution of HGF and TSG-6 in MSC-antifibrotic effects, rats were treated with MSCs silenced for HGF or TSG-6. HGF and TSG-6 silencing in transplanted MSCs resulted in a significant increase in ECM deposition in colon. These results emphasize the potential of MSCs to influence the pathophysiology of fibrosis-related diseases, which represent a challenging area for innovative treatments.

## 1. Introduction

Fibrosis appears in most tissues further to an injury. These pathologies greatly impact the quality of life of the patients, and their prevalence is high, particularly in developed countries. The potential number of patients who will suffer from colorectal fibrosis in the future may have been underestimated. The surgical removal of the fibrotic tissue is often required and is associated with a high morbidity/mortality, particularly in the case of colon and rectum diseases. The colon and rectum are susceptible to radiation-induced fibrosis and intestinal bowel diseases (IBDs), such as ulcerative colitis (UC) and Crohn’s disease (CD), which are often features of fibrosis [1].

Similar to IBDs, radiation can induce colorectal damage such as severe mucosal damage, chronic inflammation and activation of pro-fibrotic genes. Chronic insults, such as those induced by persisting inflammation, prevent proper tissue regeneration and continuously activate repair pathways. These injuries lead to differentiation of resident and recruited myofibroblasts and the smooth muscle cells (SMCs) of the muscular layer in a pro-fibrotic phenotype by shifting from a contractile phenotype to a secretory phenotype [2,3]. Transforming growth factor beta (TGF-β)/small mother against decapentaplegic (SMAD) pathway is a major contributor to fibrogenic process [4]. Concomitant upregulation of the extra cellular matrix components (ECM), ECM-degrading enzymes (matrix metalloproteinases, MMPs) and their inhibitors; tissue inhibitors of metalloproteinases (TIMPs) causes a defect in ECM turnover [5]. Late phase of colorectal fibrosis is characterized by accumulation of ECM, which occasionally infiltrates the muscle layers, and a paucicellular microenvironment due to ischemia and mechanical stress [2,6]. 

The number of new cancer cases worldwide in 2020 was 19,292,789 cases, with 9,958,133 deaths [7]. Targeted cancer therapies may be more therapeutically beneficial for lung cancer, colorectal cancer, breast cancer, lymphoma and leukemia. Targeted molecular therapy, such as against programmed cell death ligand-1 (PD-L1) and genes from the neurotrophin tyrosine kinase receptor (NTRK) family, will probably reduce the amount of radiotherapy and will improve the consequence for cancer patients [8]. The abdomino-pelvic area is home to high-incidence cancers (e.g., prostate, colorectal and cervix), currently 60% of which are treated with ionizing radiations, often associated with chemotherapy and/or surgery [9]. An estimated 20% of patients suffer from late complications up to 20 years after pelvic radiotherapy [10]. One of the common features for the treatment of pelvic radiotherapy disease (PRD) comprising fibrosis is the fact that the treatment is mostly palliative. Indeed, especially in case of fibrosis no efficient curative treatment currently exists in spite of the availability of numerous candidate therapeutic drugs, including anti-inflammatory molecules and drugs that target TGF-β signaling pathway [1]. In contrast, a large number of preclinical and clinical studies suggest that cell therapy can inhibit fibrosis development in various organs, such as the heart, liver, kidneys and gut (reviewed in [1]. Mesenchymal stem cells (MSCs) have been identified to secrete specific anti-fibrotic proteins. Notably, HGF and TSG-6 have proven to be major effectors of the anti-fibrotic effects of MSCs in several models (e.g., skin and kidney fibrosis) [11,12]. However, despite the proven beneficial effects of MSC therapy on radiation enteritis [13,14], there is slight evidence of anti-fibrotic effects of MSCs in colorectal fibrosis [15]. Furthermore, the mechanisms underlying these effects are unknown. 

Based on the need to better manage colorectal fibrosis, we aimed to investigate mediators and mechanisms that are involved in protection of MSCs against pelvic radiotherapy-induced fibrosis. An in vivo Sprague–Dawley rat’s model of colorectal fibrosis using irradiation was implemented. We assessed the effects of injection of rat MSCs on colorectal fibrosis. We set up an in vitro model of human irradiated cocultured of MSCs in presence of the main pro-fibrotic cells in colon which are fibroblasts or smooth muscle cells irradiated. Then the role of HGF and TSG-6 as effectors of the anti-fibrotic effects of MSCs on colorectal fibrosis was explored on pro-fibrotic cells. Finally, to confirm the involvement of HGF and TSG-6 in MSC-mediated effects, rats were injected with MSCs that were silenced for either HGF or TSG-6.

## 2. Results

### 2.1. In Vivo Model of Colorectal Irradiation-Induced Fibrosis

Sprague–Dawley rats were exposed to a single dose of 29 Gy, which was delivered to a 2 × 3 cm colorectal (Figure 1A), in order to obtain severe radiation-induced epithelial alterations histologically. According to a previous report, in the above-mentioned model, lesions were also similar to those seen in patient treated with radiotherapy and who develop colorectal fibrosis [16]. In this model, chronology of fibrosis consisted in initiation of fibrosis (4 weeks), intermediate phase (5 weeks) and phase of established fibrosis (7 weeks) after irradiation. To characterize fibrogenesis in rats after colorectal irradiation, we first described changes in the expression profile of ECM-related genes (i.e., collagens, MMPs, TIMPs, etc.) using real-time quantitative PCR as follows: ECM components (collagen and secreted protein acidic and rich in cysteine, SPARC), MMPs (ECM-degrading enzymes, MMP-2,9,12, 14), serpine-1, Serpine Family H Member (SERPINH1) and TIMPs (MMP inhibitors, TIMP-1,2, 3, 4). SPARC influences fibrotic collagen deposition. Serpine-1 gene codes for the plasminogen activator inhibitor-1 (PAI-1) which is associated with fibrosis. The results are compared to the “Control” group (neither irradiated nor injected with MSCs). 

During initiation of fibrosis, four weeks post-irradiation, ECM components collagen III and SPARC were overexpressed compared to the Control group (Figure 1A). Serpine-1 (gene coding for PAI-1) was overexpressed 4 weeks after irradiation. PAI-1 is known to be significantly elevated in fibrotic tissues [1]. Genes encoding MMPs (2, 9, 12, 13 and 14) and TIMPs (1, 2, 3 and 4) were upregulated. 

Chronic inflammation and fibrosis are often associated with specific immune infiltration. Activated macrophage markers nitric oxide synthase 2 (NOS2), arginine 2 (Arg2) were upregulated, 4 weeks post-irradiation compared to those in control rats (Figure 1B). Moreover, Gata-3 and Gata-3/T-bet (T-BoX transcription factor) ratio, which is indicative of Th2/Th1 ratio in T-helper (Th) cells, indicated a higher Th2 immune response (Figure 1B). Seven weeks post-irradiation, mucosal ulceration, oedema in submucosa stimulated by ECM accumulation, and overall thickening were observed (black arrows, Figure 1C,D). Irradiated animals showed a significant increase in ECM deposition in colon and rectum compared to control rats (Figure 1E) ECM and tissue thickening (Figure 1F).

To conclude, we described fibrosis-related features of irradiated colon–rectum area in Sprague–Dawley rats. After irradiation, an overall imbalance in components that regulate ECM turnover (i.e., ECM components, MMPs and TIMPs) might be related to an accumulation of ECM.

### 2.2. MSCs Inhibit Fibrogenesis Following Colorectal Irradiation

We have investigated whether MSC treatment may modulate the different phases of fibrogenesis after colorectal irradiation. We based our treatment procedures on previous reports from preclinical and clinical therapeutic protocols that employed exogenous MSCs to treat radiotherapy side effects and symptoms associated with an overdose of radiotherapy during prostate cancer treatment [13,15]. Rats were injected twice weekly with a quantity of 5 million of MSCs via tail vein according to our previous studies [7,17,18]. To potentiate the effect of MSCs, we performed repeated intravenous injection of MSCs before initiation of fibrosis, at 2 and 3 weeks after irradiation. As described in Figure 2A, to investigate effect of MSC therapy on radiation-induced fibrogenesis, rats were euthanized during first phase of fibrosis (4 weeks), intermediate phase (5 weeks) and lastly during stage of established fibrosis (7 weeks) after irradiation. The irradiated rats without the MSC transplantation (“Irradiated” group) were compared to animals receiving MSC injections (“Irradiated + MSCs group”).

Irradiation induced an upregulation of ECM-related genes (Figure 2B, red dots), which was prevented by transplantation of MSCs (Figure 2B, blue dots). Four weeks after irradiation, Irradiated + MSCs group displayed a significant downregulation of pro-fibrotic gene connective tissue growth factor (CTGF), which was accompanied by inhibition of collagen I, collagen III, fibronectin and SPARC expression (Figure 2B, blue dots). Five weeks after irradiation, MMP-2 concentration in the colon–rectum area of Irradiated + MSCs group was significantly increased (Figure 2C, blue dots) compared to that in the control (Figure 2C, white dots) and Irradiated group (Figure 2C, red dots). Seven weeks after irradiation, Sirius Red staining of paraffin-embedded colon tissue confirmed a significant reduction in ECM deposition in the colon–rectum area of Irradiated + MSCs group, particularly in submucosa (Figure 2E, dark arrows) compared to Irradiated group (Figure 1C, dark arrows). This observation was confirmed by measuring surface of fibrosis in longitudinal tissue sections. Seven weeks after ECM deposition, which was dramatically increased in the irradiated group (Figure 2D, red dot), compared to control (Figure 2D, white dots) and was reduced in the Irradiated + MSCs group (Figure 2D blue dots). These results show potent inhibitory effects exerted by MSCs on radiation-induced colorectal fibrosis through downregulation of ECM components and a more efficient control over ECM turnover.

### 2.3. MSCs Act Directly on Fibroblasts and SMCs to Inhibit Their Pro-Fibrotic Phenotype

We compared effects of MSCs on human Colonic Smooth Muscle Cells (hCoSMCs) and human Intestinal Fibroblasts (hIFs) that are responsible of colon fibrosis as described in Figure 3A. The role of HGF and TSG-6 as effectors of the anti-fibrotic effects of MSCs on colorectal fibrosis was explored. Influence of unirradiated MSCs (with or without knockdown of HG or TSG-6 genes) was tested on irradiated hIFs and irradiated hCoSMCs cultures using a Transwell system that allowed molecular exchanges without cell-to-cell contact. Expression in irradiated hIFs and irradiated hCoSMCs cocultured with MSCs (irradiated + hMSCs) was compared to irradiated hIFs and irradiated hCoSMCs. Expression of the pro-fibrotic TGF-β/SMAD pathway, a major contributor to the fibrogenic process and ECM components, ECM degrading enzymes (MMPs) and their inhibitors (TIMPs) were measured. The results in Figure 3B,C show the genes for which significant variations were measured. This experiment is in favor of a direct effect of MSCs on irradiated hCoSMCs and irradiated hIFs. When co-cultured with MSCs, irradiated hIFs revealed a marked reduction in expression of genes encoding TGF-β1, α-smooth muscle actin (α-SMA) and collagen III, compared with irradiated hIFs alone (Figure 3B, blue dots). Similarly, MSCs cocultured with irradiated hCoSMCs downregulated TGF-β1, collagen I and fibronectin gene expression compared with irradiated hCoSMCs alone (Figure 3C, blue dots). Expression in irradiated hIFs and irradiated hCoSMCs cocultured with MSCs silenced for HGF (irradiated + hMSCs siHGF) or silenced for TSG-6 (irradiated + hMSCs siTSG-6) was compared to irradiated hIFs and irradiated hCoSMCs. HGF silencing in MSCs lead to an overexpression of collagen III and fibronectin in irradiated hCoSMCs (Figure 3C, brown (siHGF) and green (siTSG-6) dots). TSG-6 silencing in MSCs induced upregulation of α-SMA, collagen III and fibronectin gene expression in irradiated hIFs (Figure 3B, green dots). The results are in favor of the anti-fibrotic potential of MSCs. We confirmed that both HGF and TSG-6 may play a role in MSC-mediated inhibition of colorectal-fibrosis in vitro. 

### 2.4. MSCs Limit Proliferation of ECM-Producing Cells

The action of MSCs on intestinal fibroblasts to inhibit fibrosis was investigated in vivo. During fibrosis in the colon and rectum, subepithelial myofibroblasts that are located inside mucosa and SMCs inside muscularis mucosa and muscularis propria are responsible for synthesis of pathological ECM. To determine whether inhibition of ECM accumulation by MSCs was related to suppressive effect of pro-fibrotic cells, amount of myofibroblasts was quantified into mucosa. The results are compared to those from the Control group. Expression of α-SMA, which is a main marker of differentiated myofibroblasts, was decreased in colon–rectum area of in Irradiated + MSCs group, four weeks after irradiation (Figure 4A, blue dots), suggesting a reduction in myofibroblast proliferation. Four and 5 weeks after irradiation, MSC injection significantly reduced density of α-SMA positive cells in mucosa in the colon–rectum area (Figure 4D–E, blue dots) compared to that in the Irradiated group, (Figure 4D–E, red dots). Interestingly, while there was a high number of α-SMA positive cells in mucosa (Figure 4C, dark arrows) in the Irradiated group, submucosa showed no evidence of myofibroblastic differentiation, and α-SMA expression was restricted to blood vessels despite a marked increase in ECM deposition. MSCs also suppressed overexpression of insulin-like growth factor 1 (IGF-1), which is a major regulator of SMC maturation, and regulates their ECM secretory properties in the Irradiated + MSCs group, (Figure 4B, blue dots). MSCs, thus, appear to reduce ECM production by inhibiting activation of pro-fibrotic cells after colorectal irradiation.

### 2.5. MSCs Delay Fibrosis by Favoring Th2/M2 Response 

In previous studies, we have shown the regenerative effect of MSCs colorectal damages induced by radiation exposure in pigs [13] and rats [14,16,17,18]. In these studies, MSCs appear to act through immunomodulation [13,14,18]. Based on these studies, we have investigated whether a treatment with MSCs may modulate MSC Th2 and M2 responses as mechanisms to reduce fibrosis in the colon after irradiation. Macrophages and T cells are involved in fibrosis development through various mechanisms, potentially regulating fibrogenesis. Principally, the reports have revealed that MSCs could act on fibrosis by orienting polarization of macrophages and differentiation of CD4+ T cells [13]. The results are compared to those from the Control group. During the initiation phase of fibrosis, at four weeks after irradiation, an increase in alternatively activated macrophages (M2) was measured by evaluating arginase-1 transcripts (Arg-1) in Irradiated + MSCs group, (Figure 5A, blue dots). In Irradiated group, a decrease in proportion of M2 macrophages was quantified on paraffin-embedded slides that were stained with fluorescent antibodies directed against macrophage M2 marker CD206 (Figure 5D,E red dots). In irradiated + MSCs group, M2 polarization was increased (Figure 5B, blue dots) compared to that in Irradiated group (Figure 5B, red dots). Five weeks after irradiation, M2 promotion appeared to persist as mannose receptor gene (Mrc-1), which is expressed by M2 macrophages, was overexpressed in the Irradiated + MSCs group (Figure 5B, blue dots) compared to the Irradiated group (Figure 5B, red dots). Since T-helper type 2 response supports alternative activation of macrophages, Gata3/T-bet ratio of gene expression was evaluated. A higher ratio in the Irradiated + MSCs group (Figure 5C blue dots) compared to the Irradiated group was measured indicating a shift towards Th2 differentiation in T cells (Figure 5C red dots). MSCs may promote a burst in Th2/M2 cells, which might be necessary to effective degradation of pathological ECM. These cells have previously been associated with anti-inflammatory and ECM remodeling activity, particularly during fibrosis. Data showed that MSC injection might promote M2 and Th2 polarization finally reducing fibrosis in colon. The MSC-mediated regeneration in the colon–rectum area induced a significant improvement in the animals’ survival. While 30% of the irradiated rats died before day 50 (Figure 5E red line), there was a 100% survival rate during the same time period in the MSC group (Figure 5F blue line). The reduction in the ECM deposition could, thus, ameliorate survival, most notably by restoring tissue functionality and preventing occlusion.

### 2.6. HGF Silencing in Transplanted-MSCs May Promotes Fibrosis through a TGF-β-Dependent Mechanism

Finally, in order to confirm the involvement of HGF and TSG-6 in MSC-induced effects against colorectal fibrosis, rats were injected with MSCs that were silenced for HGF (Irradiated + MSCs HGF group) or TSG-6 (Irradiated + MSCs TSG-6 group). The effect of HGF silencing was presented here, then in the following paragraph the effect of TSG-6 silencing. We retained to focus effect on established fibrosis. For this reason, we measured consequence of transplanted MSCs silencing for HGF in Irradiated + MSCs siHGF group at 7 weeks after irradiation. To check the effect of silencing, we compared Irradiated + siHGF (or siTSG-6) group with Irradiated group. HGF is a potent anti-fibrotic protein with pleiotropic activities, including immunosuppression and inhibition of TGF-β signaling. Because HGF has been shown to mediate anti-fibrotic effects of MSCs, RNA interference was used to knockdown HGF gene in MSCs prior to their injection in irradiated rats as described in Figure 6A. The knockdown of HGF in transplanted MSCs (Irradiated + MSCs siHGF) suppressed inhibition of TGF-β and ECM-related genes. Seven weeks after radiation exposure, TGFβ-1, TGFβ-2, TGFβ-3 and CTGF were upregulated in the colon–rectum area of animals transplanted with HGF-silenced MSCs (Irradiated + MSCs siHGF, Figure 6B, brown dots) compared to those in irradiated + MSCs group (blue dots). Concomitantly, collagen I, collagen III, fibronectin and spark (PAI-1), MMP-3, 9, 12, 13, TIMP-1, serpine-1 were overexpressed (Figure 6B,C, brown dots) in Irradiated + MSCs siHGF group. The stimulation of pro-fibrotic mechanisms resulted in an accumulation of pathological ECM, to a level comparable to that observed in non-transplanted irradiated rats. Thus, HGF that was secreted by MSCs in Irradiated + MSCs siHGF group prevented TGF-β mediated activation of fibrogenic pathways and consequent fibrosis as illustrated in Irradiated + MSCs siHGF group (Figure 6D brown dots) in comparison of Irradiated + MSCs group (Figure 6D blue dots).

### 2.7. TSG-6 May Inhibit Fibrosis by Deregulating Macrophage Polarization

By silencing TSG-6 in the MSCs, the TSG-6 role in suppressing colon fibrosis was investigated. Effect of transplanted MSCs silencing for TSG-6 was evaluated on an established fibrosis, 7 weeks after irradiation. TSG-6 protein is produced during inflammation. TSG-6 is involved in hyaluronan-induced leucocyte recruitment. Thus, immunomodulatory and anti-fibrotic properties of MSCs have been previously partially associated with their production of TSG-6. A TSG-6 knockdown in MSCs by RNA interference was performed (Irradiated + MSCs TSG-6). In irradiated + MSCs siTSG-6 group an upregulation of IGF-1 and collagen I and III (Figure 7A, green dots) was evidenced. Since TSG-6 protein was described as able to influence macrophage polarization, as observed in our model, proportion of M2 macrophages in the Irradiated + MSCs TSG-6 group was quantified (Figure 7B). In the Irradiated + MSCs TSG-6 group a significant increase in proportion of M2 macrophages (Figure 7B, green dots) was evidenced compared to Irradiated + MSCs group (Figure 7C, blue dots). Finally, ECM deposition was significantly increased in the colon–rectum area of Irradiated + MSCs—TSG-6-treated rats (Figure 7C, green dots) compared to that in Irradiated + MSCs group (Figure 7C, blue dots). Similar to HGF, TSG-6 secretion by MSCs could be necessary for anti-fibrotic effects.

## 3. Discussion

Colorectal fibrosis is a feature of many chronic inflammatory disorders, including Crohn’s disease and severe chronic complications of pelvic radiotherapy. Efficient therapy for these disorders has not been set up so far. MSCs exert various anti-fibrotic effects in human and animal models. MSCs have regenerative and anti-inflammatory properties that have been used to treat inflammatory pathologies, including graft-versus-host disease (GvHD) [19] and radiological burns [20]. In this report, our results imply that MSCs limit activation of pro-fibrotic cells, particularly myofibroblasts, SMCs and macrophages, by releasing HGF and TSG-6.

Our in vivo study confirms the anti-fibrotic effect of MSCs by controlling ECM turnover. The in vitro results suggest that HGF and TSG-6 act via different mechanisms on myofibroblasts and SMCs. MSCs that are silenced TSG-6 appears to have a selective effect on hIFs and MSCs silenced HGF have a selective effect on HCoSMCs. MSCs that are silenced for HGF induced a strong upregulation of collagen III and fibronectin in HCoSMCs. TSG-6 silencing in MSCs may induce an upregulation of myofibroblasts marker α-SMA and genes encoding collagen III and fibronectin. To settle these results, it would be necessary to validate the involvement of HGF and TSG-6 by a neutralizing antibody study. In agreement with previous studies on fibrosis, we observed that HGF and TSG-6 may exert complementary function necessary for anti-fibrotic effects of MSCs during radiation-induced fibrosis [21,22,23].

Our in vivo study shows that, following irradiation, a dense population of α-SMA positive sub-epithelial myofibroblasts was evidenced, which likely accounted for ECM deposition in mucosa. α-SMA expression was restricted to blood vessels nevertheless to confirm that the α-SMA increase corresponds to an increase in capillary formation, a CD31 labelling might be necessary. MSC transplantation, induced a decrease of myofibroblast population that could explain lower ECM gene expression. Moreover, the colon–rectum area of the MSC-treated animals exhibited a lower CTGF expression. CTGF prolongs fibrogenesis during chronic stage of radiation-induced fibrosis [24]. Indeed the downregulation of CTGF could, thus, reduce pro-fibrotic signals in tissue, including myofibroblastic differentiation [1]. Furthermore, thickening of the muscularis propria and ECM deposition surrounding muscle fibers after irradiation support involvement of pro-fibrotic SMCs that we observed in accordance with a previous study [14]. We have shown that the most important effect of the HGF silencing is the upregulation of genes encoding TGF-β, ECM components, MMP and TIMP. The consequence was an in favor of an increased fibrosis. Our results are in accordance with a previous study which demonstrated that MSCs that overexpress HGF produce improved anti-fibrotic effects on lung and liver fibrosis compared to wild-type MSCs [21,22,23]. This report puts forward that MSCs secrete HGF, which may be necessary to maintain low TGF-β signaling, resulting in inhibition of fibrogenesis as previously reported in other models of fibrosis [1].

Similarly, TSG-6 silencing in transplanted MSCs resulted in an increase in ECM deposition in the colon and rectum. This result is in agreement with previous studies which establish that TSG-6 is involved in regulation of pro-inflammatory and pro-fibrotic hyaluronan activity [25,26]. TSG-6 secreted by MSCs could participate in regulation of macrophage activity. MSC treatment induced a significant increase in M2/M1 ratio of macrophage polarization, which was supported by Th2 immune response. In agreement with previous reports, we hypothesize that MSCs might limit fibrosis, by modulating macrophage polarization [1]. 

The impact of HGF and TSG-6 silencing may highlight importance of secretion of a wide range of factors by therapeutic MSCs. Compared to pharmacological treatments, MSC therapy offers advantage of providing a pleiotropic response by activating multiple mechanisms instead of acting on a single pathway. 

## 4. Materials and Methods 

We retained the use of human cells to identify key effectors secreted by MSCs on cell types which induced colon fibrosis. Then, the results were validated in a rat model of colon fibrosis. 

### 4.1. In Vitro Experiments

A co-culture system was implemented to investigate influence of MSCs on intestinal fibroblasts and colonic smooth muscle Cells that are responsible for colon fibrosis. Human MSCs were cultured in a minimal essential medium α containing 20% FBS, 1% penicillin/streptomycin and 1% L-glutamine, which were obtained from Thermo Fisher Scientific (Illkrich, France). Cells were used between passages 2 and 5. 

Primary human colonic smooth muscle cells (hCoSMCs, ref: 2940) and human intestinal fibroblasts (hIFs, ref: 2920-SC) were obtained from ScienCell (6076 Corte Del Cedro, Carlsbad, CA 92011, USA) and cultivated, respectively, in smooth muscle Growth Medium-2 (SmGM-2™, ref: CC-3182) and Fibroblasts Growth Medium-2 (FGM-2™, ref: CC-3132), which were obtained from Lonza Verviers (Verviers, Belgium). 

In summary, 6000 MSCs/cm^2^ were seeded on Transwell^®^ chambers with 0.4 µm pores (Corning Life Sciences) and hIFs or hCoSMCs were seeded on 6-well plates (Falcon) at a density of 6000 and 5000 cells/cm^2^, respectively. After 4 days, hIFs and hCoSMCs were irradiated with a single dose of 20 Gy (2.43 Gy/min) to induce pro-fibrotic phenotype. After 12 hours, MSCs (upper part of transwell) were transferred to hIF and hCoSMC cultures. HIFs and hCoSMCs were collected 12 hours after transfer, and dry cell pellets were stored at −80 °C prior to RNA extraction. Additional details regarding this protocol are provided in Figure 3A.

### 4.2. RNA Interference against HGF and TSG-6

HGF and TSG-6 are major effectors of anti-fibrotic effect of MSCs as evidenced by both in vitro and in vivo experiments. HGF and TSG-6 genes were silenced in MSCs in both in vitro co-culture and in vivo transplantation experiments. ON-TARGETplus SMARTpools of siRNAs, which were designed to improve knockdown efficiency, and other transfection reagents were purchased from GE Healthcare Sa (Vélizy-Villacoublay, France). SiRNAs used in these experiments were specific for following genes: human HGF (reference L-006650-00-0050), rat HGF (reference L-089896-02-0050), human TSG-6 (ref: L-012379-00-0050) and rat TSG-6 (ref: L-108341-00-0050). The transfection of human and rat MSCs was performed as instructed by manufacturer. In brief, human or rat MSCs were seeded at a density of 3000 cells/cm^2^. Cells were transfected at 80% confluence with a solution containing 2 µL/mL Dharmafect1 (ref: T-2001-03) and 1 µM of siRNA in a complete MSC growth medium (as described before). A preliminary experiment was performed to assess silencing efficiency. RNA interference against GAPD gene was used as a positive control (ON-TARGETplus GAPD Control Pool, ref: D-001830-10-05), and a non-targeting siRNA served as a negative control (ref: D-001810-10-05). mRNA was extracted from these cells, and expression levels of HGF and TSG-6 were quantified using RT-qPCR. In in vitro and in vivo experiments, MSCs were used 48 hours after transfection, when silencing was most effective. The silencing of HGF and TSG-6 genes is effective over a period of one week.

### 4.3. In Vivo Experiments

All experiments were implemented in agreement with French laws and guidelines for animal experiments (Act no. 92–333 of 2 October 2009) and approved by the Ethics Committee of Animal Experimentation “CEEA number 810” (Protocol numbers: P09–10). In total, 104 Sprague–Dawley (SD) male rats (300 g) were purchased from Janvier Labs (LeGenest St Isle, France). eGFP transgenic rats (strain “green rat CZ-C04 Tg Act eGFP”) derived from Sprague–Dawley strain were obtained by the Institute for Radiological Protection and Nuclear Safety (IRSN) from Pr. Otabe (Osaka University, Osaka, Japan) with MTA and subsequently bred in the IRSN’s animal housing facility. The progeny of the eGFP rats were systematically tested for expression of transgene. Transgenic rats were used as the source of the GFP-labelled MSCs. The animals were housed in double-decker cages (two or three per cage) with ad libitum access to food and water and light and dark cycles. All efforts were made to minimize suffering, and all experiments were performed under gaseous anesthesia with isoflurane (Aerrane, Baxter SA, Lessines, Belgium). Animal behavioral and physiological parameters (e.g., bleeding and diarrhea) were monitored daily, and suffering animals were euthanized. Euthanasia was performed in a CO_2_ chamber. Sprague–Dawley rats, aged ten weeks (250–300 g) at the beginning of experiment, were divided into 2 batches. The Control batch consisted of 6 animals that were neither irradiated nor injected with MSCs (“Control” group). Other batch underwent colorectal irradiation, followed by transplantation of MSCs. In total, 18 animals were irradiated without the MSC transplantation (“Irradiated” group), and 18 animals received injections with wild-type MSCs (“Irradiated + MSCs” group). Additional details of protocol are shown in Figure 2A. Finally, two groups of 6 animals received GFP-MSCs that were silenced for either HGF (“Irradiated + MSCs siHGF” group) or TSG-6 (“Irradiated + MSCs siTSG-6” group). Additional details are provided in Figure 6A.

In another experiment, 24 rats were irradiated, and 26 rats were irradiated and injected with MSCs according to the protocol described above to determine effect of the MSCs on survival of the irradiated rat (Figure 5F).

### 4.4. Isolation, Characterization and Culture of MSCs

GFP-MSCs were extracted from adipose tissue obtained from seven-week-old eGFP transgenic rats as previously described [14]. Briefly, subcutaneous inguinal adipose tissue was removed from eGFP-SD rats, finely minced and enzymatically digested with 0.1% collagenase type I (reference: C0130, Sigma-Aldrich, St Quentin Fallavier, France) at 37 °C, and then filtered through a 30 µm filter. This process was repeated three times, and remaining collagenase was neutralized with 10% Foetal Bovine Serum (FBS) in extract. Cells were washed with Phosphate Buffered Saline (PBS) and suspended in MEM-α containing 20% FBS, 1% Penicillin/Streptomycin and 1% L-Glutamine. Cells were seeded at 1000 cells/cm^2^, and medium was changed after 4 days. On day 7, a monolayer of adherent cells was trypsinized (trypsin-0.25% EDTA, Thermo Fisher Scientific), washed three times with PBS and suspended in 10 millions of cells/mL. Phenotype of amplified Ad-MSCs was verified by flow cytometry. Presence of CD90 (clone OX-7; BD Biosciences, Le Pont de Claix, France) and CD73 (clone 5F/B9; BD Biosciences) was assessed in culture, and absence of haematopoietic lineage cells was verified with CD34 (clone ICO115; Santa Cruz Biotechnology, Inc., Dallas, USA) and CD45 (clone OX-1; BD Biosciences) markers. Controls were performed using isotype identical antibodies. Adipogenic, osteogenic and chondrogenic differentiation capacity of MSCs was evaluated as described by Rochefort et al. 25. The ability to form Colony-Forming Unit Fibroblasts (CFU-F) was also analysed. CFU-Fs were stained with crystal violet and counted 10 days after initial seeding.

### 4.5. Configuration of Irradiation, MSC Treatment and Tissue Sampling

Rats were exposed to an X-ray source (2.43 Gy/min) and received a single radiation dose of 29 Gy as previously described [14]. Radiation field was confined to a 2 × 3 cm window that was localized to colon–rectum area. According to our previous studies, the groups of rats treated (“Irradiated + MSCs” group) were injected twice weekly with quantity of 5 million of adipose MSCs via the tail vein. Injections occurred before fibrosis at 2 and 3 weeks after irradiation during initiation of fibrosis. To investigate the effect of the MSC therapy on the radiation-induced fibrogenesis, rats were euthanized during the first phase of fibrosis (4 weeks), intermediate phase (5 weeks) and the phase of established fibrosis (7 weeks) after the irradiation. Rats that received MSC-siHGF or MSC-siTSG-6 (“Irradiated + MSCs siHGF” group and “Irradiated + MSCs siTSG-6” group) were euthanized 7 weeks after the irradiation. Colon and rectum were sampled to evaluate the progression of the radiation-induced lesions (e.g., mucosal damage and ECM deposition). Additional details regarding this protocol are provided in Figure 2A and Figure 6A. Euthanasia was performed under isoflurane anesthesia. After euthanasia, we sampled 3 cm of the colon–rectum area, which was included in the radiation field. The colon–rectum area was opened lengthwise, separated into 3 longitudinal sections and processed for future experiments. RNALater^®^ (ref: AM7024, Qiagen, Les Ulis, France) was used to preserve samples that were used for gene expression analysis. For protein quantification, samples that were used for RNA extraction were rapidly frozen in liquid nitrogen and stored at −80 °C. Final sections were immersed in 4% buffered formaldehyde, dehydrated and paraffin-embedded. Each group of animals was composed of 6 rats for a given period of time (e.g., 6 weeks). The results of each group were compared as follows: Irradiated group was compared to Control group; Irradiated + MSCs group was compared to control and Irradiated group; and Irradiated + MSCs siHGF and Irradiated + MSCs siTSG-6 groups were compared to Irradiated + MSCs group.

### 4.6. RNA Isolation, Reverse Transcription, Quantitative Real-Time PCR, and Array Analysis

Tissue and cell total RNA were extracted using RNeasy^®^ Mini Kit (ref: 74106, Qiagen) from cell cultures and RNeasy^®^ Fibrous Tissue Mini Kit (ref: 74704) from colon–rectum samples. Rat samples were first disrupted using TissueRuptor disposable probes (ref: Qiagen). Purity and concentration of RNA extracts were assessed by spectrophotometry before reverse transcription. One µg of mRNA was used to produce cDNAs for each sample using a High-Capacity cDNA Reverse Transcription Kit (ref: 4368813, Thermo Fisher Scientific). RT-qPCR was performed using TaqMan^®^ Universal PCR Master Mix (ref: 4326708) and TaqMan^®^ Gene Expression Assays (ref: 4331182), which were purchased from Thermo Fisher Scientific. QuantStudio™ 12K Flex Real-Time PCR System (ref: 4471090, Thermo Fisher Scientific) was used to evaluate gene expression profiles. Gene expression data analysis was performed using QuantStudio™ 12K Flex Real-Time PCR Software and ExpressionSuite Software 1.1 (Thermo Fisher Scientific). Relative Quantity (RQ) of each gene was calculated using the 2-ΔΔCt method. In the in vitro experiments, GAPDH served as an endogenous control, and irradiated cells were selected as reference group. The gene expression profiles of rats were assessed using Tyrosine 3-Monooxygenase/Tryptophan 5-Monooxygenase Activation Protein Zeta (Ywhaz) as an endogenous control. Gene-specific Taqman probe and primer sets were obtained from Thermo Fisher Scientific as Assays-on-Demand gene expression products. For the in vitro study, the identification numbers were Hs00998133_m1for TGF-β1, Hs00426835_g1 for α-SMA, Hs00164099_m1 for collagen I, Hs00943809_m1 for collagen III, Hs00365052_m1 for fibronectin and Hs02758991_g1 for the human GAPDH endogenous control. For the in vivo study, the identification numbers were Rn00572010_m1 for TGF-β1, Rn00676060_m1 for TGF-β2, Rn00565937_m1 for TGF-β3, Rn01526721_m1 for collagen I, Rn01437681m1 for collagen III, Rn00692663_m1 for fibronectin, Rn01470624_m1 for SPARC, Rn01759928 for MMP-2, Rn00579162m1 for MMP-9, Rn01448194_m1 for MMP-13, Rn00579172_m1 for MMP-14, Rn01481341_m1 for Serpine-1, Rn00567777_m1 for Serpinh1, Rn01430873_g for TIMP-1, Rn00573232_m1 for TIMP-2, Rn00441826_m1 for TIMP-3, Rn01459160_m1 for TIMP-4, Rn00710306_m1 for IGF-1, Rn00583646_m1 for akt1, Rn00591471_m1 for ilk and Rn00755072_m1 for ywhaz the rat endogenous control.

### 4.7. Protein Extraction, Quantification and ELISAs

Tissue samples that were stored at −80 °C were thawed and finely minced prior to protein extraction. Proteins were extracted by tissue disruption using TissueRuptor (Qiagen) in ice-cold PBS containing protease inhibitors (Complete Mini, Roche SAS, Boulogne-Billancourt, France). After centrifugation, protein concentration was assessed in supernatants using Pierce™ BCA Protein Assay Kit (Thermo Fisher Scientific). ELISAs for rat MMP-2 (ref: EK0639, Boster Biological Technology Co., Ltd., Pleasanton, CA, USA), MMP-9 (ref: E-EL-R0624, Elabscience Co., Ltd., Wuhan Shi, China) and TGF-β1 (ref: EK0514, Boster Biological Technology Co., Ltd.) were performed as instructed by their respective manufacturers.

### 4.8. Histology and Immunohistochemistry (IHC)

The paraffin-embedded samples were cut into 4 μm sections and stained with Hematoxylin-Eosin-Safran (HES) or PicroSirius (Direct Red 80, ref: 365548-25G, Sigma-Aldrich^®^, Lyon, France). The lengths and fibrosis areas in full-length longitudinal tissue sections were analysed. ECM areas were automatically measured using Histolab software (Microvision Instruments, Lisses, France) to automatically detect Picrosirius stained surfaces (magnification 40×). Percentage of fibrosis area was obtained by dividing ECM area by total surface of tissue in each field, which was manually measured by Histolab. For immunohistochemical protocols, paraffin-embedded sections were dewaxed, rehydrated, and then permeabilized with a solution containing 0.1% Triton (N3020-100 ML, Sigma-Aldrich) to characterize myofibroblast. The staining of α-SMA required an extra step in methanol solution containing 0.3% H_2_O_2_ to inhibit endogenous peroxidases. The antigen retrieval was performed by 3 × 5 min microwave cycles in Target Retrieval Solution (ref: S169984-2, Agilent Technologies, Les Ulis, France). Primary antibodies against α-SMA (ref: A2547, Sigma-Aldrich), CD68 (ab125212, Abcam, Paris, France) and CD206 (bs-2664R, Bioss antibodies™, Woburn, MA, USA) were used for myofibroblast and macrophage quantification. We used an HRP Horse Anti-Mouse IgG secondary antibody (ref: PI-2000, Vector Laboratories, Inc., Burlingame, CA, USA) and Histogreen (ref: E109, Linaris, Dossenheim, Germany) to reveal α-SMA staining, followed by a counterstaining with Nuclear Fast Red solution (ref: N3020, Sigma-Aldrich). For co-staining of CD68 and CD206, secondary antibodies that were conjugated with fluorescent dyes Alexa Fluor 568 (Goat anti-Rabbit, red, ref: A-11011, Thermo Fisher Scientific) and Alexa Fluor 488 (Donkey anti-Rabbit, green, ref: A-21206, Thermo Fisher Scientific) were employed, respectively to characterized M2 macrophage. The fluorescent slides were mounted with Vectashield^®^ HardSet Antifade Mounting Medium with DAPI (ref: H-1500, Vector Laboratories, Inc.) to show cell nuclei in blue under a fluorescence microscope. We used Fiji software (Fiji Is Just ImageJ, http://imagej.net/Fiji) to perform measurements on these slides. Myofibroblasts and M1 and M2 macrophages were numbered on their respective slides with 5 fields per sample (magnification: 100×), and total surface of tissue in each field was measured to calculate cell density. M1 macrophages stained positive for CD68 only (red) and negative for CD206 (green), while M2 macrophages appeared yellow because they stained positive for both CD68 and CD206. Staining was correlated with corresponding PCR results. 

### 4.9. Statistical Analysis

Statistical analyses were performed using SigmaPlot v11 (Systat Software GmbH, Erkrath, Germany). Two-group comparisons were performed with *t*-tests, while one-way ANOVAs, followed by Bonferroni *t*-tests, were used for multiple group comparisons. Survival curves were compared with log-rank test. The results are expressed as mean ± SD. A value of *p* ≤ 0.05 is considered statistically significant. In figures, asterisks correspond to *p* < 0.001 (***), *p* < 0.01 (**) and *p* < 0.05 (*).

A Log-Rank survival analysis was performed using R (R Foundation). The competitive risk methodology was used to discriminate between non-radiation-related deaths (during anaesthesia or MSC injection) and radiation-induced mortality. 

## 5. Conclusions

Fibrosis is a leading cause of death in occidental countries, and therapies are palliative. The increasing number of patients with fibrosis requires innovative approaches. We identified for the first time two key effectors that are secreted in vivo by transplanted MSCs that limited the sustained progression of colon fibrosis. We characterized molecular mechanisms of these two effectors in inflammatory cells and pro-fibrotic cells and demonstrated their differing and complementary effects. These findings have important implications for the use of radiotherapy in cancer patients and inflammatory bowel disease (IBD) and should encourage the clinical use of MSCs. These results further emphasize the potential of MSCs to influence the pathophysiology of fibrosis-related diseases, which represent a challenging area for future research. 

## Figures and Tables

**Figure 1 ijms-22-01790-f001:**
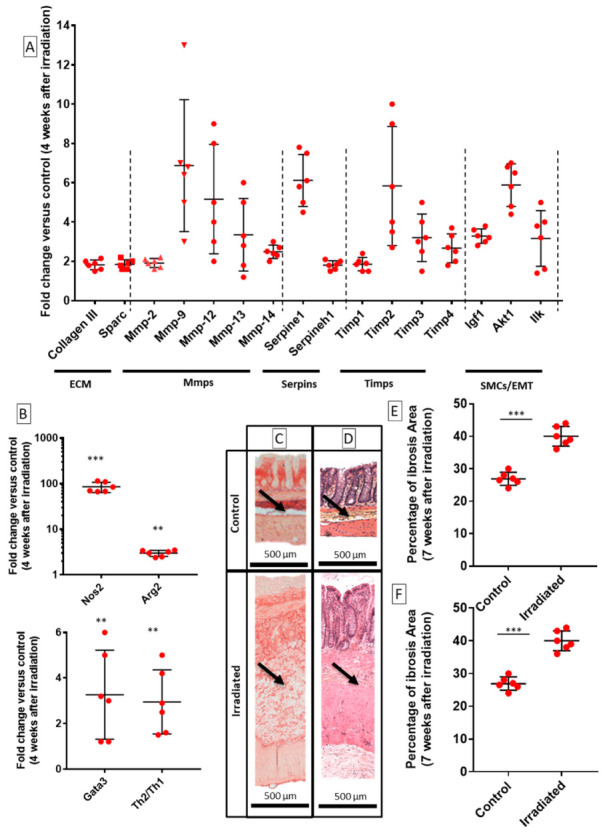
Features of radiation-induced colorectal fibrosis model in Sprague–Dawley rats. (**A**) mRNA fold changes of genes 4 weeks after irradiation that are normalized to Ywhaz and standardized against control (neither irradiated nor injected with mesenchymal stem cells (MSCs)). Extra cellular matrix (ECM) components, matrix metalloproteinases (MMPs), serpine-1 (PAI-1) and TIMPs were upregulated in colon–rectum area of irradiated rats. (**B**) mRNA fold changes of macrophage and T-cell response. (**C**) Representative histological images of colon of control and irradiated rats showing ECM (red) stained with Picrosirius Red (origin magnification: 40×). Seven weeks after irradiation, animals display oedema in submucosa that was caused by accumulation of ECM. (**D**) Representative images of colon of control and irradiated rats stained with HES 7 weeks after irradiation (original magnification: 40×). Black arrows indicate ECM accumulation compared to control. (**E**) Fibrosis deposition in the colon–rectum area of control and irradiated animals 7 weeks after irradiation. Percentage of fibrosis area quantified on Picrosirius Red-stained slides showing a significant increase in ECM deposition after irradiation. (**F**) Irradiation induces a significant thickening of tissue, particularly in submucosa and muscularis propria, 7 weeks after irradiation. Each group of animals was composed of 6 rats. Results were compared as follows: Irradiated group was compared to Control group. Two-group comparisons were performed with *t*-tests, while one-way ANOVAs, followed by Bonferroni *t*-tests, were used for multiple group comparisons. The results are expressed as mean ± standard deviation (SD). **: *p* < 0.01; ***: *p* ≤ 0.001.

**Figure 2 ijms-22-01790-f002:**
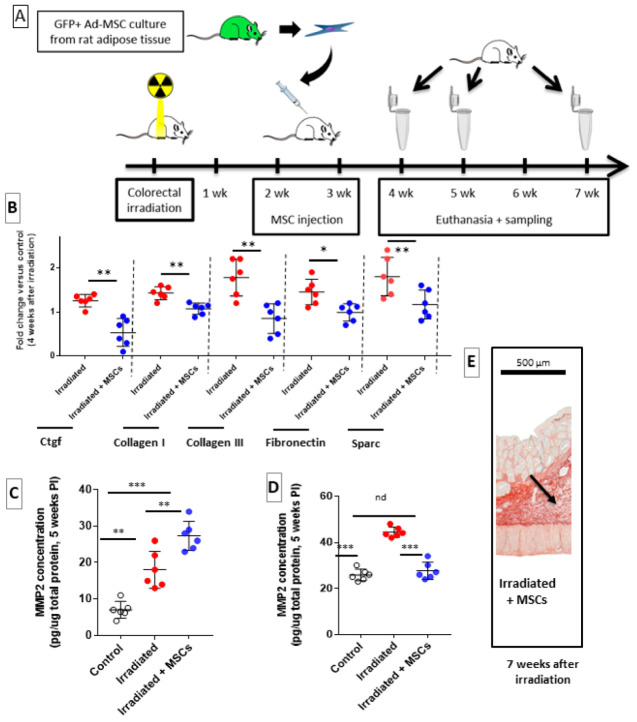
Effect of MSC transplantation on radiation-induced ECM remodeling. (**A**) Timeline of irradiation and MSC transplantation protocol. (**B**) mRNA fold changes of genes 4 weeks after irradiation (red dots) and 1 week after MSC injection (blue dots) that are normalized to Ywhaz and standardized against Control group. mRNA fold changes of genes encoding CTGF and ECM components indicate suppression of pro-fibrotic signals by MSCs. (**C**) Five weeks after irradiation: MSCs increase secretion of MMP-2 in the irradiated colon–rectum area. (**D**) Percentage of fibrosis area quantified on Picrosirius Red-stained slides. MSCs significantly inhibit ECM deposition in the colon–rectum area of irradiated animals as observed 7 weeks after irradiation. (**E**) Representative images showing ECM (black arrow) stained with Picrosirius red (original magnification: 40×) in the colon–rectum area of irradiated + MSCs group, 7 weeks after irradiation (original magnification: 40×). Each group of animals was composed of 6 rats. The results of each group were compared as follows: Irradiated group was compared to Control group; Irradiated + MSCs group was compared to control and Irradiated group. Two-group comparisons were performed with *t*-tests, while one-way ANOVAs, followed by Bonferroni *t*-tests, were used for multiple group comparisons. The results are expressed as mean ± SD. *: *p* < 0.05; **: *p* < 0.01; ***: *p* ≤ 0.001.

**Figure 3 ijms-22-01790-f003:**
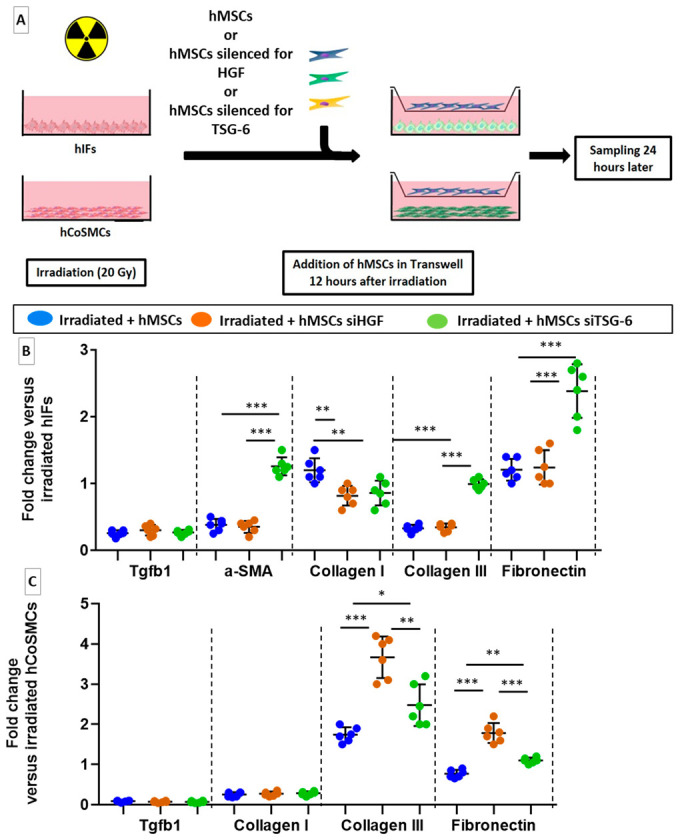
In vitro experiments show a direct effect of human MSCs and their effectors, HGF and TSG-6, on pro-fibrotic activity of irradiated hIFs and irradiated hCoSMCs. (**A**) hIFs and hCoSMCs were irradiated with a single dose of 20 Gy, and MSCs or MSCs silenced for HGF or TSG-6 were added to Transwell chambers 12 h after irradiation. (**B**,**C**) mRNA fold changes in fibrosis-related genes 24 h after MSC addition that are normalized to GAPDH and standardized against irradiated hIFs or irradiated hCoMSCs. (**B**) MSCs reduced common features of pro-fibrotic gene expression in irradiated hIFs and (**C**) irradiated hCoSMCs. (**B**) Silencing of TSG-6 in MSCs may counteract effect on α-SMA, collagen III and fibronectin expression in irradiated hIFs. (**C**) Silencing HGF in MSCs may counteract effect on collagen III and fibronectin expression in irradiated hCoSMCs. Two-group comparisons were performed with *t*-tests, while one-way ANOVAs, followed by Bonferroni *t*-tests, were used for multiple group comparisons. The results are expressed as mean ± SD. *: *p* < 0.05; **: *p* < 0.01; ***: *p* ≤ 0.001.

**Figure 4 ijms-22-01790-f004:**
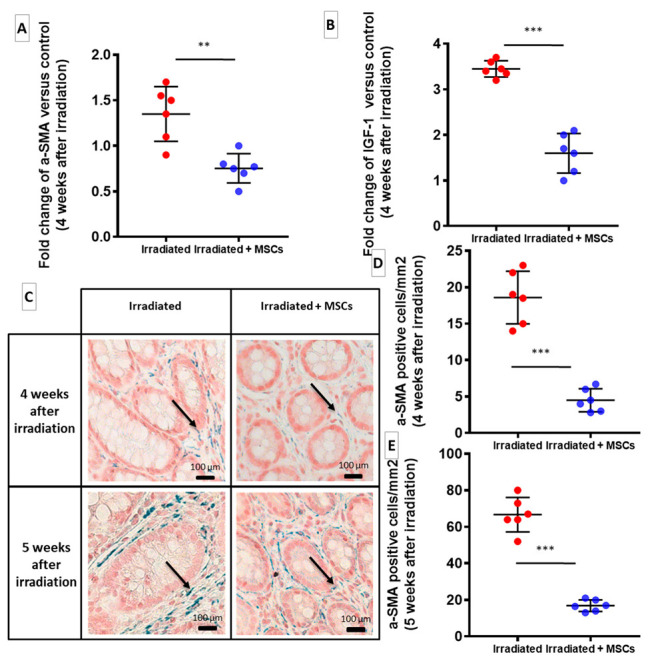
Involvement of myofibroblasts and SMCs in radiation-induced fibrosis and anti-fibrotic effects of MSCs. (**A**,**B**) mRNA fold changes of α-SMA and IGF-1, 4 weeks after irradiation and 1 week after the last MSC injection, which are normalized to Ywhaz and standardized against the Control group. (**A**) MSCs inhibit activation of α-SMA-positive myofibroblasts. (**B**) MSCs decrease expression of IGF-1, a potent activator of SMC’s pro-fibrotic phenotype. (**C**) Representative histological images of paraffin-embedded slides of colon–rectum samples stained for α-SMA (black arrows): 4 weeks and 5 weeks after irradiation (Irradiated: left; Irradiated + MSCs: right; original magnification: 100×). (**D**,**E**) Quantification of α-SMA-positive cells in mucosa of colon–rectum area of Irradiated groups with or without MSC therapy. MSCs significantly decrease number of α-SMA-positive cells in mucosa of irradiated rats 4 weeks (**D**) and 5 weeks (**E**) after irradiation. Each group of animals was composed of 6 rats. The results of each group were compared as follows: Irradiated group was compared to Control group; Irradiated + MSCs group was compared to control and Irradiated group. Two-group comparisons were performed with *t*-tests, while one-way ANOVAs, followed by Bonferroni *t*-tests, were used for multiple group comparisons. The results are expressed as mean ± SD. **: *p* < 0.01; ***: *p* ≤ 0.001.

**Figure 5 ijms-22-01790-f005:**
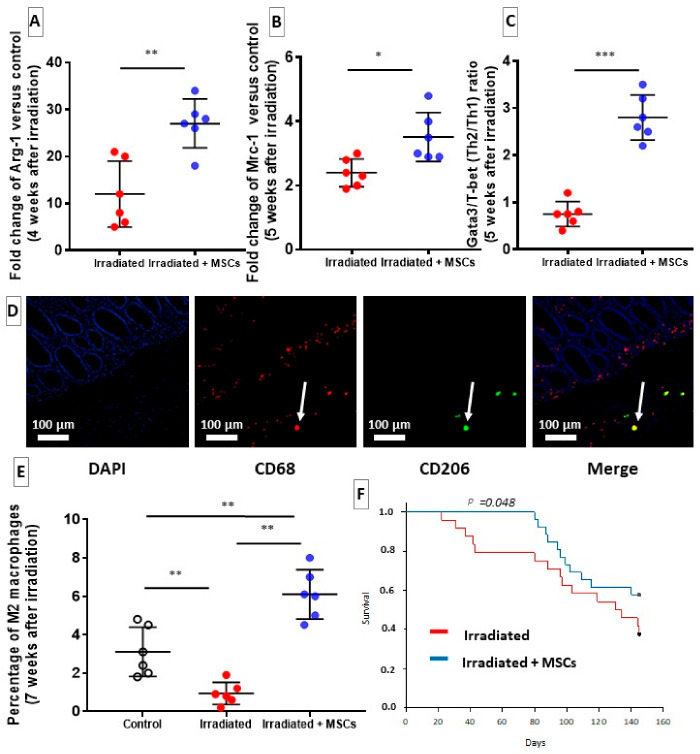
Effect of MSC transplantation on macrophage polarization in irradiated colon–rectum area. (**A**,**B**) mRNA fold changes of M2 markers Arginase-1 and Mannose Receptor-1 that are normalized to Ywhaz and standardized against control. (**C**) MSCs inverted Th2/Th1 balance in favor of Th2 cells 5 weeks after irradiation, as shown by increased Gata3/T-bet ratio gene expression. (**D**) M2 polarization was identified by co-staining paraffin-embedded tissue slides with 4′,6-diamidino-2-phenylindole (DAPI) (cell nuclei, blue), CD68 antibody (red) and CD206 antibody (green). M2 macrophages are shown in yellow (merge) (original magnification: 100×). MSC injection induced a strong M2 polarization of macrophages 4 weeks after irradiation. (**E**) MSCs favor M2 polarization compared to control and Irradiated group. (**F**) Survival curve of Irradiated group (red line, n = 24) and Irradiated + MSCs group (blue line, n = 26). Risk comparison analysis showed a significant increase in survival after MSC transplantation (*p* = 0.048). Each group of animals was composed of 6 rats. Results of each group were compared as follows: Irradiated group was compared to Control group; Irradiated + MSCs group was compared to control and Irradiated group. Two-group comparisons were performed with *t*-tests, while one-way ANOVAs, followed by Bonferroni *t*-tests, were used for multiple group comparisons. Survival curves were compared with the log-rank test. The results are expressed as mean ± SD. *: *p* < 0.05; **: *p* < 0.01; ***: *p* ≤ 0.001.

**Figure 6 ijms-22-01790-f006:**
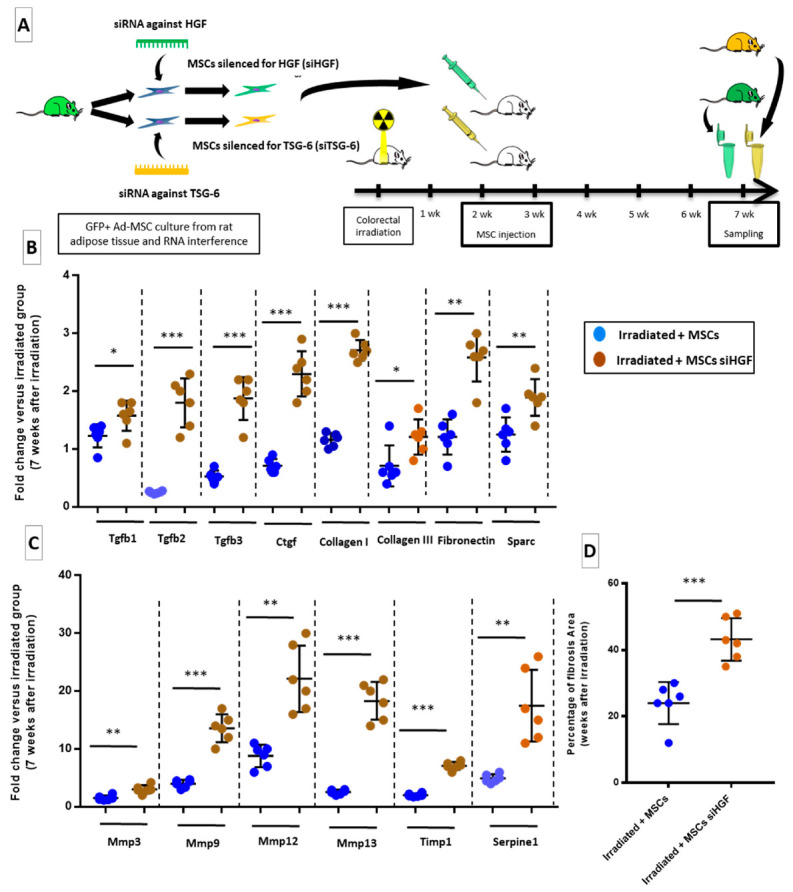
HGF silencing in transplanted MSCs induces pro-fibrotic signals in irradiated rats and stimulates ECM production (7 weeks after irradiation) (**A**) Protocol: two distinct groups of rats were injected with MSCs that were silenced for HGF and TSG-6 genes. (**B**,**C**) mRNA fold changes of genes 7 weeks after irradiation that are normalized to Ywhaz and standardized against irradiated group. MSCs silenced for HGF induce a significant remodeling in the colon–rectum area of irradiated rats, mainly by activating TGF-β signaling. (**B**) mRNA fold changes in genes encoding TGF-β family and ECM components. (**C**) mRNA fold changes in genes encoding MMPs, TIMP-1 and serpine-1 (PAI-1). (**D**) Percentage of fibrosis area quantified on Picrosirius Red-stained paraffin-embedded slides show a significant increase in ECM deposition after transplantation of MSCs silenced for HGF. Each group of animals was composed of 6 rats. The results of each group were compared as follows: Irradiated + MSCs siHGF and Irradiated + MSCs siTSG-6 groups were compared to Irradiated + MSCs group. Two-group comparisons were performed with *t*-tests, while one-way ANOVAs, followed by Bonferroni *t*-tests, were used for multiple group comparisons. The results are expressed as mean ± SD. *: *p* < 0.05; **: *p* < 0.01; ***: *p* ≤ 0.001.

**Figure 7 ijms-22-01790-f007:**
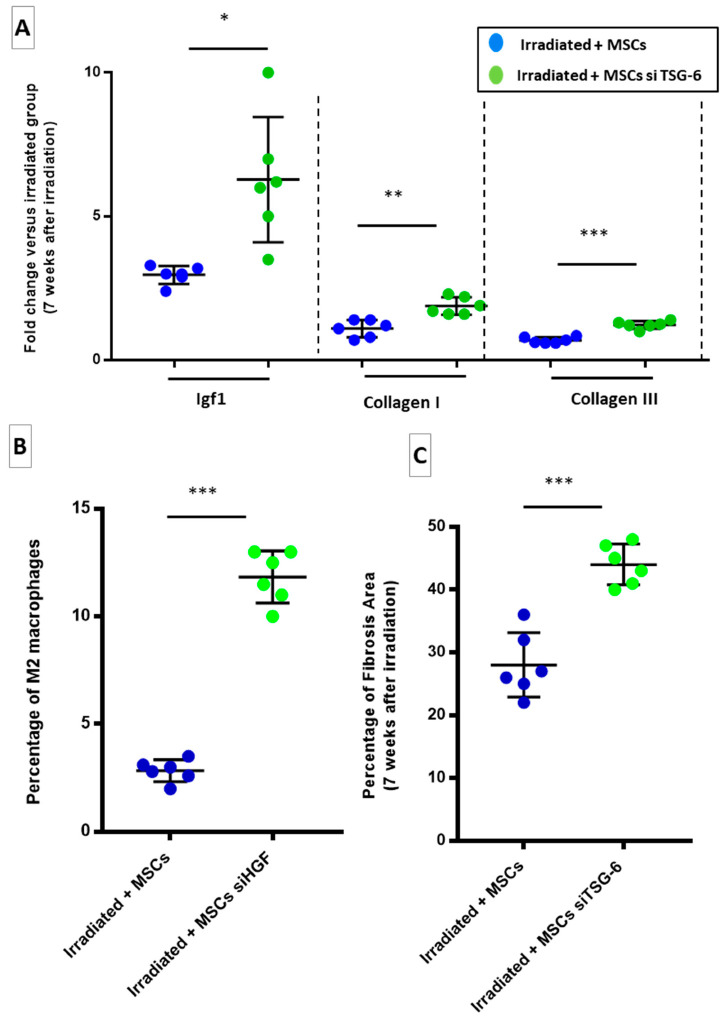
TSG-6 silencing in transplanted MSCs induces pro-fibrotic signals in irradiated rats and stimulates ECM production (7 weeks after irradiation). (**A**) mRNA fold changes in fibrosis-related genes 7 weeks after irradiation that are normalized to Ywhaz and standardized against irradiated group. MSCs silenced for TSG-6 induce a strong remodeling in colon–rectum area of irradiated rats, mainly by collagen produced by SMCs activated by IGF-1. (**B**) Percentage of M2 macrophages 7 weeks after irradiation (4 weeks after MSC injection) as identified by CD68-CD206 co-staining. MSCs silenced for TSG-6 stimulate chronic activation of pro-fibrotic M2 macrophages. (**C**) Percentage of fibrosis area quantified on Picrosirius Red-stained paraffin-embedded slides show a significant increase in ECM deposition after transplantation of MSCs silenced for TSG-6. Each group of animals was composed of 6 rats. The results of each group were compared as follows: Irradiated + MSCs siHGF and Irradiated + MSCs siTSG-6 groups were compared to Irradiated + MSCs group. Two-group comparisons were performed with *t*-tests, while one-way ANOVAs, followed by Bonferroni *t*-tests, were used for multiple group comparisons. The results are expressed as mean ± SD. *: *p* < 0.05; **: *p* < 0.01; ***: *p* ≤ 0.001.

## Data Availability

The data that support the findings of this study are available from the corresponding author upon reasonable request.

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
