# Peer review of "HGF and TSG-6 Released by Mesenchymal Stem Cells Attenuate Colon Radiation-Induced Fibrosis"

_ijms, 2021, doi:10.3390/ijms22041790_

Round 1

Reviewer 1 Report

In the paper entitled “HGF and TSG-6 released by mesenchymal stem cells attenuate colon 
radiation-induced fibrosis”, Benoit Usunier and colleagues described the healing potential of

MSC in reducing the fibrosis induced by radiation.

The concept of a beneficial effect of MSC in radiation-induced fibrosis is not new (see Moussa et al, 2016), but the paper well describes the anti-fibrotic effects of MSC, with a presumptive role of HGF and TSG-6, in colorectal fibrosis.

COMMENTS

The in vitro experiments should be located after Section 2.3

The involvement of both HGF and TSG-6 should be evaluated in vitro using neutralizing antibodies.

MSC silenced for Hgf or Tsg-6 are just described in the M&M section. Please show the details of their characterization.

In Fig. 1 the control of MSC supernatant vs non-irradiated fibroblasts is missing.  

Please provide in the text the rational for choosing the genes evaluated in Fig. 1 B and C.

Panel A in Figure 2 is wrong: MSC are not used in these experiments, please remove them from the panel. The panel containing MSC goes on Figure 3.

Statistics of Figure 2B is missing.

Does the increase in aSMA correspond to an increase in capillary formation? Provide CD31 staining as well.

Please provide explanation of aSMA induction in IF.

I’m not convinced by the statistics. It’s impossible that in fig. 6 B the tgfb2 difference is **, while the b3 is ***. Please provide the Delta CT data and the relative statistics for all the expression panels

MINOR

In the legend of Fig. 1 panel A, I suggest inserting the delta symbol between MSC and the deleted gene. The same in Fig. 6 and 7. In panel C a Col3a2 (?) gene is mentioned.

R145 Please change “responsible”. There is no demonstration of responsibility in the data.

R155 Provide description of HES

In Figure 3 Panel A, in Figure 4 panels A and B, in Figure 5 panels A,B and C, the legend names the control, but from the results the “control” seems to be the “Irradiated” group. Please provide explanation.

R227 Please control figure legend: A and B are mentioned twice

R297 and following. There is a lot of confusion between 7 weeks or 4 weeks after irradiation

TYPOS

In Fig. 2 there are two C and two F panels

R 96, 99, 108, 110 Abbreviation of Col3a1 was given already in raw 93

R 124 collage (?)

R 134 Dot is missing

R 143 Finally (?)

R 151 Remove extra space after response

R156/158 Please exchange G and F letters

From R168 Please control reference style

R174 Remove extra space after MSC. A space is missing after +. Hyphen are missing after MSC

R177 and R180 A space is missing after +

Figure 3A: legends are in french, not in english

R236 please control

R237 Please remove “that”

R240 Substitute By/With

R245 Change “activated macrophage was measured (M2) Arginase-1 transcripts (Arg-1) in Irradiated + MSCs” in “activated macrophages (M2) was measured by evaluating Arginase-1 transcripts (Arg-1)”.

R248 Insert “directed against” before macrophage M2

R249 … compared to the control group… and to the

R259 Might promote

R308 TSG-6 role in suppressing

R309 Please remove “at”

R325 Insert space between Figure and 7. Again 7 weeks and 4 weeks confusion

Author Response

Dear Editors and reviewers,

Thank you for your efforts in reviewing our manuscript entitled " HGF and TSG-6 released by mesenchymal stem cells attenuate colon radiation-induced fibrosis” (Manuscript No: ijms-1059739). We truly appreciate the in-depth criticisms and constructive comments of the reviewers. Overall, the referees are enthusiastic about the significance of our work. However you expressed some concerns, they are mostly about our unclear course of the article. In the rewrite, following your advice we tried to find a satisfactory solution. We have addressed the questions raised by the reviewers and some additional data are added. We are grateful for these valuable comments and feel that after incorporating the reviewers’ advice, the revised manuscript has been significantly strengthened and that the significance of our findings is now more evident. A detailed point-by-point response to reviewers’ comments is attached. Changes to the original manuscript are marked by underlined text.

Thank you for your time and effort. We look forward to hearing your decision.

Sincerely yours,

Dr Alain CHAPEL

IRSN PSE SANTE SERAMED LRMED

PO17 92262 FAR france

Response to Reviewers’ Comments

Reviewer #1:

We are grateful that the reviewer believes that “the paper well describes the anti-fibrotic effects of MSC, with a presumptive role of HGF and TSG-6, in colorectal fibrosis”. We have carefully considered the reviewer’s comments and advice and performed new data and revised our manuscript accordingly. We believe that these changes have further strengthened the manuscript.

The concept of a beneficial effect of MSC in radiation-induced fibrosis is not new (see Moussa et al, 2016).

Thank you for noting the publication of Lara Moussa and Benoît Usunier. We have cited this outstanding publication which improves our article.

Line 67: However, despite proven beneficial effects of MSC therapy on radiation enteritis (12, 13), there is slight evidence of anti-fibrotic effects of MSCs in colorectal fibrosis (14).

Reference 14: Bowel Radiation Injury: Complexity of the Pathophysiology and Promises of Cell and Tissue Engineering. Lara Moussa, Benoît Usunier, Christelle Demarquay, Marc Benderitter, Radia Tamarat, Alexandra Sémont, Noëlle Mathieu. Cell Transplant . 2016 Oct;25(10):1723-1746. doi: 10.3727/096368916X691664.

  1. The in vitro experiments should be located after Section 2.3

Thank you for this excellent suggestion, in vitro experiments were located in section 2.3

This the reason why, the article was re-organized and in vitro experiments were located after Section 2.3. In order to respect the logic of the results, this reorganization required rewriting a number of paragraphs in the summary, introduction and discussion. The modified parts are highlighted.

  1. The involvement of both HGF and TSG-6 should be evaluated in vitro using neutralizing antibodies.

We thank the reviewer for this excellent suggestion, however to our regret we cannot add these results because the experiment has not been done and an is not feasible in the short term.

For this reason we mentioned in the discussion that it would be necessary to validate the involvement of HGF and TSG-6 in vitro by neutralizing antibodies.

Lines 365-6: In discussion, we added : “To settle these results, it would be necessary to validate the involvement of HGF and TSG-6 by a neutralizing antibody study.”

  1. MSC silenced for Hgf or Tsg-6 are just described in the M&M section. Please show the details of their characterization.

Find details of characterization of MSC silenced for Hgf or Tsg-6 48 hours after transfection:

A preliminary experiment was performed to assess silencing efficiency.

mRNA were extracted from these cells, and expression levels of HGF and TSG-6 were quantified using RT-qPCR. In in vitro and in vivo experiments, MSCs were used 48 hours after transfection, when silencing was effective. Results are expressed as 2- DD Ct

1) Relative expression of HGF in hMSCs HGF or TSG-6 in hMSCs TSG-6 compared to untreated MSCs.

2) Relative expression of HGF in hMSCs HGF or TSG-6 hMSCs TSG-6 compared to untreated MSCs.

Relative expression of HGF or TSG-6 (48hrs)

hMSC HGF

rMSC HGF

hMSC TSG-6

rMSC TSG-6

Fold Change

0,09

0,08

0,07

0,09

% compared to un-transfected MSCs

9,43%

8,37%

7,26%

8,53%

HGF and TSG-6 mRNA levels were decreased by >90% in the silenced MSCs, 48 hours after transfection.

  1. In Fig. 1 the control of MSC supernatant vs non-irradiated fibroblasts is missing.

Results are expressed in fold change compared to the control of irradiated cells in the absence of MSC. Introducing non-irradiated controls therefore requires the addition of figures. However, the fold change compared to the same cells not irradiated in the presence of MSC has not been introduced because the results are insignificant.

Explanation are below:

The calculated relative expression of a given target in irradiated (hCoSMCs or hIF) after cocultured with MSC was calculated according to previously described formula.

DD Ct = D Ct target (CT sample-Ct control)- D Ct reference (Ct sample - Ct control).

Results are expressed as 2- DD Ct

This calculation was based on the expression ratio of a target gene versus a reference gene (GAPDH). The relative expression ratio of a target gene was computed, based on the crossing point difference (DCt) of an unknown sample (irradiated cocultured (hIF or hCoSMC with MSC) versus a control (hIF or hCoSMC irradiated alone). DCt (Ct control - CT sample) obtained for a target gene was normalized on the crossing point difference (DCt) obtained from the reference gene (GAPDH).

Here, for TGFb, DD Ct = DCt target (Ct TGFb-Ct GAPDH)- DCt reference (Ct TGFb - Ct GAPDH).

                                               hIF iiradiated cocultured MSC                   -   hIF iiradiated alone

Results were expressed as a fold increase (>1) or decrease (<1) in the mRNA level of target genes in irradiated cells hIF or hCoSMC co-cultured with MSC as compared to the control (irradiated hIF or hCoSMC cells alone).

For this reason, if we introduced non irradiated cells, we need to introduce supplementary figures without significative results.

Reference: M.W. Pfaff, G.W. Horgan, L. Dempfle, Relative expression software tool (REST) for group-wise comparison and statistical analysis of relative expression results in real-time PCR. Nucleic Acids Res. 30 (2002) e36.

Fold change of MSC vs non-irradiated hIF or hCoSMC compared to non-irradiated hIF or hCoSMC alone.

Gene name

TGFB1

a-SMA

Collagen I

Collagen III

Fibronectin

hIFs

Fold change

0,98

1,01

0,89

1,10

1,08

SEM

0,03

0,11

0,12

0,06

0,15

hCoSMCs

Fold change

1,11

1,03

1,15

0,97

1,04

SEM

0,15

0,07

0,05

0,05

0,13

Line 177, we added “Expression in irradiated hIFs and irradiated hCoSMCs cocultured with MSCs (irradiated + hMSCs) was compared to irradiated hIFs and irradiated hCoSMCs.”

Line 187 we added “Expression in irradiated hIFs and irradiated hCoSMCs cocultured with MSCs silenced for HGF (irradiated + hMSCs HGF) or silenced for TSG-6 (irradiated + hMSCs TSG-6) was compared to irradiated hIFs and irradiated hCoSMCs.”

  1. Please provide in the text the rational for choosing the genes evaluated in Fig. 1 B and C.

Thank you for this excellent suggestion which improves the understanding of the results. Concerning “the rationale for choosing these genes” we have added the following;

Lines 179-82: The expression of genes involved in fibrosis have been studied. Expression of pro-fibrotic TGF-β/SMAD pathway, a major contributor to the fibrogenic process and ECM components, ECM degrading enzymes (MMPs) and their inhibitors (TIMPs) were measured. The results in figures 1b and 1c shown the genes for which significant variations were measured.

  1. Panel A in Figure 2 is wrong: MSC are not used in these experiments, please remove them from the panel. The panel containing MSC goes on Figure 3.

Thank you for pointing out this error and we apologize, see (Line 159).

  1. Statistics of Figure 2B is missing.

Thank you for pointing out this error, we added statistic in Figure 2B (now Figure 1B)

  1. Does the increase in aSMA correspond to an increase in capillary formation? Provide CD31 staining as well.

We thank the reviewer for this excellent suggestion, however to our regret we cannot add these results because the experiment has not been done and is not feasible in the short term.

Lines 372-74: We mentioned in the discussion that “α-SMA expression was restricted to blood vessels nevertheless to confirm that a-SMA increase correspond to an increase in capillary formation, a CD31 labelling might be necessary.”

  1. Please provide explanation of aSMA induction in IF.

As we added above, to explain α-SMA induction in irradiated group, we mentioned that this increase might be related an increase of blood vessels.

Lines 372-74: We mentioned in the discussion that “α-SMA expression was restricted to blood vessels nevertheless to confirm that the α-SMA increase corresponds to an increase in capillary formation, a CD31 labelling might be necessary.”  

  1. I’m not convinced by the statistics. It’s impossible that in fig. 6 B the tgfb2 difference is **, while the b3 is ***. Please provide the Delta CT data and the relative statistics for all the expression panels

We would like to thank the reviewer for detecting this error that occurred during the assembly of the figures. We apologize for the inconvenience. According to reviewer’s comment we checked the statistic.

MINOR

  1. In the legend of Fig. 1 panel A, I suggest inserting the delta symbol between MSC and the deleted gene. The same in Fig. 6 and 7. In panel C a Col3a2 (?) gene is mentioned.

Figure 1A (now 2 A) is the methodological scheme, if I understand the comment well, I guess that the delta symbol has to be introduced figure 1B and C? But we don't understand where it has to be introduced in relation to the deleted gene (I guess it is the silenced gene). Similarly for figures 6 and 7, we assume that it is figures 6B and C, 7A. Is it a question of replacing the fold changes by delta Ct (2-ΔΔCt)? We will do it with pleasure if this point can be explained please.

  1. R145 Please change “responsible”. There is no demonstration of responsibility in the data

Lines 108 to 109: we change sentence “After irradiation, an overall …was responsible for accumulation of ECM.”

Line 109: “After irradiation, an overall imbalance in components that regulate ECM turnover (i.e., ECM components, MMPs and TIMPs) might be related to an accumulation of ECM.

  1. R155 Provide description of HES

In legend of figure 2, for HES; Line 20, we added sentence “Black arrows indicate ECM accumulation compared to control.”

  1. In Figure 3 Panel A, in Figure 4 panels A and B, in Figure 5 panels A,B and C, the legend names the control, but from the results the “control” seems to be the “Irradiated” group. Please provide explanation.

In figure 5 and 6, the results are expressed in relation to the irradiated group in order to appreciate the effect of silencing HGF or TSG-6. We have checked at the level of the paragraphs of the corresponding results that there are no more errors or contradictions.

To discard any confusion; we added:

Line 212 and 246: “Results are compared to those from the Control group.” (Figure 4 and 5);

Line 298: “To check effect of silencing, we compared Irradiated + HGF (or TSG-6) group to Irradiated group.”

In this case, results was expressed as:

DD Ct = DCt target (Ct TGFb1-Ct Ywaz)- DCt reference (Ct TGFb1 - Ct Ywaz).

            Irradiated + MSCs (or Irradiated + MSCs HGF or TSG-6)         -   Irradiated group

Reference gene (Ywaz)

  1. R227 Please control figure legend: A and B are mentioned twice

We modified the sentence in legend of Figure 4 by:

Line 232 “(A,B) mRNA fold changes of α-SMA and IGF-1, 4 weeks after irradiation and 1 week after the last MSC injection, which are normalized to Ywhaz and standardized against irradiated group.”

  1. R297 and following. There is a lot of confusion between 7 weeks or 4 weeks after irradiation

We apologize for this mistake, sentence was changed for:

Line 316: “(B, C) mRNA fold changes of genes 7 weeks after irradiation that are normalized to Ywhaz and standardized against irradiated group.”

  1. TYPOS

We checked Typos in the whole article (modifications are highlighted).

  1. In Fig. 2 there are two C and two F panels

Corrected, in Figure 2, thank you

  1. R 96, 99, 108, 110 Abbreviation of Col3a1 was given already in raw 93

We removed Col3a1 in article and replaced by collagen III, thank you.

  1. R 124 collage (?)

Line 88 (now); changed for collagen

  1. R 134 Dot is missing

Line 97 (now). Dot is added; “….upregulated.“

  1. R 143 Finally (?)

Line 107 (now): Changed for “To conclude, we described…”

  1. R 151 Remove extra space after response

Line 115 now; extra space is removed

  1. R156/158 Please exchange G and F letters

Line 120: Changed, for the sentence; “(E) Fibrosis deposition in the colon-rectum area of control and irradiated animals 7 weeks after irradiation. Percentage of fibrosis area quantified on Picrosirius Red-stained slides showing a significant increase in ECM deposition after irradiation. (F) Irradiation induces a significant thickening of tissue, particularly in submucosa and muscularis propria, 7 weeks after irradiation.”

  1. From R168 Please control reference style

Line 133 now. Modified, “……….previous studies (7, 17, 18).”

  1. R174 Remove extra space after MSC. A space is missing after +. Hyphen are missing after MSC

Line 141 (now). Thank you, corrected for “..animals receiving MSC injections (“Irradiated + MSCs group”).”

  1. R177 and R180 A space is missing after +

Line 145. Irradiated + MSCs group

  1. Figure 3A: legends are in french, not in English

Thank you, we corrected legend in Figure 3A

  1. R236 please control

Line 244. Changed for “2.5. MSCs delay fibrosis by favouring Th2/M2 response”

  1. R237 Please remove “that”

Line 245. Changed for “In previous studies, we have shown the regenerative effect of MSCs colorectal damages induced by radiation exposure in pigs (12) and rats (13, 15-17).”

  1. R240 Substitute By/With

Line 248. Changed for “whether a treatment with MSC may modulate MSCs Th2 and M2”

  1. R245 Change “activated macrophage was measured (M2) Arginase-1 transcripts (Arg-1) in Irradiated + MSCs” in “activated macrophages (M2) was measured by evaluating Arginase-1 transcripts (Arg-1)”.

Line 252. We modified this sentence, according to your suggestion, thank you for this correction.

“During the initiation phase of fibrosis, at four weeks after irradiation, an increase in alternatively activated macrophages (M2) was measured by evaluating Arginase-1 transcripts (Arg-1) in Irradiated + MSCs group, (Figure 5A, blue bar).”

  1. R248 Insert “directed against” before macrophage M2

Line 256. Sentence was changed “……..with fluorescent antibodies directed against macrophage M2 marker CD206 (Figures 5D, 5E red bar).”

  1. R249 … compared to the control group… and to the

Line 260. “which is expressed by M2 macrophages, was overexpressed in the Irradiated + MSCs group (Figure 5B, blue bar) compared to the Irradiated group (Figure 5B, red bar).”

(Control group is removed, see comments to reviewer 2)

  1. R259 Might promote

Line 267. Data showed that MSC injection might promote M2 and Th2 polarization finally reducing fibrosis in colon.

  1. R308 TSG-6 role in suppressing

Line 326. Changed for “………..By silencing Tsg-6 in MSCs, Tsg-6 role in suppressing colon fibrosis…”

  1. R309 Please remove “at”

Line 326. Changed for “…..Effect of transplanted MSC silencing for Tsg-6 was evaluated on an established fibrosis, 7 weeks after”

  1. R325 Insert space between Figure and 7. Again 7 weeks and 4 weeks confusion

Line 327 “Figure 7. TSG-6 silencing in”

Reviewer 2 Report

ijms-1059739

General Comments

This manuscript seems to describe the significance of mesenchymal stem cells with altered HGF and TSG-6 expression after the irradiation. However, it is hard to read because of illogicality and redundancy. Please remove unnecessary text from the manuscript. In addition, numerous typos and errors are found throughout the manuscript. The extensive revision would be required before publication.

Major Points

  1. Line 54 Although the authors described the estimated ratio of radiation-induced colonic fibrosis, the prevalence of such fibrotic disease in general population or number of newly diagnosed cases per year is unclear. Please provide the epidemiological data.

  1. Line 90-111 Explanatory texts for Figure 1B and 1C are difficult to understand. Please revise them concisely and clearly. Please also summarize the difference between HGF and TSG6 or hIFs and hCoSMCs.

  1. Figures 2D and 2E. The representative images of control and irradiated colon seem to not be the identical magnification. What does the scale bars mean in Figure 2D and 2E, respectively?

  1. Figure 2F seems to be duplicated.

  1. The order of the data appeared in the manuscript seems to be peculiar. Please consider it.

  1. Figures 3. The representative images of control and irradiated + MSCs colon seem to not be the identical magnification. What does the scale bars mean?

  1. Line 250. No white bar is found in Figure 5B.

  1. Line 426. “3x103 cells/cm2” seems to be odd.

  1. In the quantitative real-time PCR analysis, there is no primer information. Please provide it.

Author Response

Dear Editors and reviewers,

Thank you for your efforts in reviewing our manuscript entitled " HGF and TSG-6 released by mesenchymal stem cells attenuate colon radiation-induced fibrosis” (Manuscript No: ijms-1059739). We truly appreciate the in-depth criticisms and constructive comments of the reviewers. Overall, the referees are enthusiastic about the significance of our work. However you expressed some concerns, they are mostly about our unclear course of the article. In the rewrite, following your advice we tried to find a satisfactory solution. We have addressed the questions raised by the reviewers and some additional data are added. We are grateful for these valuable comments and feel that after incorporating the reviewers’ advice, the revised manuscript has been significantly strengthened and that the significance of our findings is now more evident. A detailed point-by-point response to reviewers’ comments is attached. Changes to the original manuscript are marked by underlined text.

Thank you for your time and effort. We look forward to hearing your decision.

Sincerely yours,

Dr Alain CHAPEL

IRSN PSE SANTE SERAMED LRMED

PO17 92262 FAR france

Response to Reviewers’ Comments

 Reviewer #2:

We thank the reviewer for the thorough review and helpful advice. We have carefully considered the reviewer’s comments and advice and revised our manuscript accordingly. We carefully checked on the integrity of the article for typo and errors.

  1. However, it is hard to read because of illogicality and redundancy. Please remove unnecessary text from the manuscript. In addition, numerous typos and errors are found throughout the manuscript. The extensive revision would be required before publication.

We thank the reviewer for this suggestion. We have removed repetitions into results and discussion sections, numerous typos and errors in the revised manuscript. The missing part has been added. We apologize for having provided you with an article with these defects.

Reviewer 1 asked to move in vitro results in 2.3 (instead of 2.1), in order to respect the logic of the results, this reorganization required rewriting a number of paragraphs in the summary, introduction and discussion. The modified parts are highlighted.

  1. Line 54 although the authors described the estimated ratio of radiation-induced colonic fibrosis, the prevalence of such fibrotic disease in general population or number of newly diagnosed cases per year is unclear. Please provide the epidemiological data

This is a good point; we thank the reviewer for this suggestion. Radiation-induced colonic fibrosis, in general population is just estimated. What we know is number of newly diagnosed cases of cancer per year and we know that 60% of cancer are treated by radiotherapy. About 20% of patient treated by radiotherapy suffer from late complications, including fibrosis. We have not the precise number of patients developing fibrosis after radiotherapy.

Accordingly, we have revised the manuscript and added following paragraph:

Line 54 to 57. “The number of new cancer cases worldwide has been continuously increasing during the past decades and is expected to reach 19.3 million in 2025 (7). The abdomino-pelvic area is home to high-incidence cancers (e.g. prostate, colorectal and cervix), 60% of which are treated with ionizing radiations, often associated with chemotherapy and/or surgery (8).”

  1. Ferlay J, Soerjomataram I, Dikshit R, Eser S, Mathers C, Rebelo M, et al. Cancer incidence and mortality worldwide: sources, methods and major patterns in GLOBOCAN 2012. International journal of cancer. 2015;136(5):E359-86. Epub 2014/09/16. [cited 2016 11/18/2016]; Available from: http://globocan.iarc.fr

  1. Hauer-Jensen M, Denham JW, Andreyev HJ. Radiation enteropathy--pathogenesis, treatment and prevention. Nature reviews Gastroenterology & hepatology. 2014;11(8):470-9. Epub 2014/04/02.

  1. Line 90-111 Explanatory texts for Figure 1B and 1C are difficult to understand. Please revise them concisely and clearly. Please also summarize the difference between HGF and TSG6 or hIFs and hCoSMCs.

According to the reviewer’s recommendation we have rewritten explanatory texts for Figure 1B and 1C:

Line 176 to 202. We compared effects of MSCs on human Colonic Smooth Muscle Cells (hCoSMCs) and human Intestinal Fibroblasts (hIFs) that are responsible of colon fibrosis as described in Figure 3A. The role of HGF and TSG-6 as effectors of the anti-fibrotic effects of MSCs on colorectal fibrosis was explored. Influence of unirradiated MSCs (with or without knockdown of HG or TSG-6 genes) was tested on irradiated hIFs and irradiated hCoSMCs cultures using a Transwell system that allowed molecular exchanges without cell-to-cell contact. Expression in irradiated hIFs and irradiated hCoSMCs cocultured with MSCs (irradiated + hMSCs) was compared to irradiated hIFs and irradiated hCoSMCs. Expression of the pro-fibrotic TGF-β/SMAD pathway, a major contributor to the fibrogenic process and ECM components, ECM degrading enzymes (MMPs) and their inhibitors (TIMPs) were measured. The results in figures 3B and 3C show the genes for which significant variations were measured. This experiment highlighted a direct effect of MSCs on irradiated hCoSMCs and irradiated hIFs. When co-cultured with MSCs, irradiated hIFs displayed a marked reduction in expression of genes encoding TGF-β1, α-Smooth Muscle Actin (α-SMA) and collagen III, compared with irradiated hIFs alone (Figure 3B, blue bar). Similarly, MSCs cocultured with irradiated hCoSMCs downregulated TGF-β1, collagen I and fibronectin gene expression compared with irradiated hCoSMCs alone (Figure 3C, blue bar). Expression in irradiated hIFs and irradiated hCoSMCs cocultured with MSCs silenced for HGF (irradiated + hMSCs HGF) or silenced for TSG-6 (irradiated + hMSCs TSG-6) was compared to irradiated hIFs and irradiated hCoSMCs. HGF silencing in MSCs resulted in overexpression of collagen III and fibronectin in irradiated hCoSMCs (Figure 3C, hatched blue bars). TSG-6 silencing in MSCs induced upregulation of α-SMA, collagen III and fibronectin gene expression in irradiated hIFs (Figure 3B, dotted blue bars). Results established the anti-fibrotic potential of MSCs. We confirmed that both HGF and TSG-6 play a role in MSC-mediated inhibition of colorectal-fibrosis in vitro.

Figure 3. In vitro experiments show a direct effect of human MSCs and their effectors, HGF and TSG-6, on pro-fibrotic activity of irradiated hIFs and irradiated hCoSMCs. (A) hIFs and hCoSMCs were irradiated with a single dose of 20 Gy, and MSCs or MSCs silenced for HGF or TSG-6 were added to Transwell chambers 12 hours after irradiation. (B, C) mRNA fold changes in fibrosis-related genes 24 hours after MSC addition that are normalized to GAPDH and standardized against irradiated hIFs or irradiated hCoMSCs. (B) MSCs reduced common features of pro-fibrotic gene expression in irradiated hIFs and (C) irradiated hCoSMCs. (B) Silencing of TSG-6 in MSCs counteracts effect on α-SMA, collagen III and fibronectin expression in irradiated hIFs. (C) Silencing HGF in MSCs counteracts effect on collagen III and fibronectin expression in irradiated hCoSMCs. Results are expressed as the mean value ± SEM. *: p<0.05; **: p<0.01; ***: p≤0.001.

  1. Figures 2D and 2E. The representative images of control and irradiated colon seem to not be the identical magnification. What does the scale bars mean in Figure 2D and 2E, respectively?

Thank you for this remark. The images have been readjusted to the same size and the scale has been set for each photo. The scale bars in Figure 2D and 2E mean five hundred micrometres (now C and D). The scale bars have been added on each photo. In legend of figure magnification was indicated (original magnification: 40X).

  1. Figure 2F seems to be duplicated. (now 2 E)

We removed this duplication in Figure 2E.

  1. The order of the data appeared in the manuscript seems to be peculiar. Please consider it.

Thank you, according to the 2 reviewer’s suggestion, we have revised the order of figures and paragraph.

Line 199. Figure 1, now is figure 3

Line 171. Paragraph 2.1 becomes 2.3

We hope that this change improves the understanding of the article however if the reviewers propose another order, we will not hesitate to move the in vitro results to the place that seems best for this  understanding of the article,

  1. Figures 3. The representative images of control and irradiated + MSCs colon seem to not be the identical magnification. What does the scale bars mean?

Figure 2 (now).We removed duplication. The scale bars in Figure E mean five hundred micrometres. The scale bars have been added on photo and magnification is in legend of figure 3 (origin magnification: 40X).

  1. Line 250. No white bar is found in Figure 5B.

Line 257. We apologize for this mistake, we have revised following line.

New sentence is “In irradiated + MSCs group, M2 polarization was increased (Figure 5B, blue bar) compared to that in irradiated group (Figure 5B, red bar).”

  1. Line 426. “3x103 cells/cm2” seems to be odd.

Line 430. Thank you for noticing this error, and we apologize for the inconvenience. Sentence was rewritten “”…density of 3000 cells/cm2

  1. In the quantitative real-time PCR analysis, there is no primer information. Please provide it.

Information on the primers is now provided in section 4.6. Line 508

Gene-specific Taqman probe and primer sets were obtained from Thermo Fisher Scientific as Assays-on-Demand gene expression products. For in vitro study, the identification numbers were Hs00998133_m1for TGF-β1, Hs00426835_g1 for a-SMA, Hs00164099_m1 for collagen I, Hs00943809_m1 for collagen III, Hs00365052_m1 for fibronectin and Hs02758991_g1 for the human GAPDH endogenous control. For in vivo study, the identification numbers were   Rn00572010_m1 for TGF-β1, Rn00676060_m1 for TGF-β2, Rn00565937_m1 for TGF-β3, Rn01526721_m1 for collagen I, Rn01437681m1 for collagen III, Rn00692663_m1 for fibronectin, Rn01470624_m1 for SPARC, Rn01759928 for MMP-2 , Rn00579162m1 for MMP-9, Rn01448194_m1 for MMP-13, Rn00579172_m1 for MMP-14, Rn01481341_m1 for Serpine-1, Rn00567777_m1 for Serpinh1, Rn01430873_g for TIMP-1, Rn00573232_m1 for TIMP-2, Rn00441826_m1 for TIMP-3, Rn01459160_m1 for TIMP-4, Rn00710306_m1 for IGF-1, Rn00583646_m1 for akt1, Rn00591471_m1 for ilk and Rn00755072_m1 for ywhaz the rat endogenous control.

Round 2

Reviewer 1 Report

I believe the authors did a great job of reworking the manuscript.

Nevertheless, there is still some questionable points. My last reply is in bold.

  1. The involvement of both HGF and TSG-6 should be evaluated in vitro using neutralizing antibodies.

We thank the reviewer for this excellent suggestion, ……..

I cannot scientifically accept the statement of the involvement of TSG-6 and HGF in the inhibition of radiation-induced colonic fibrosis based on the only experiment conducted with silenced MSCs, especially for the prominence that these two molecules have been given, also found in the title.

In my opinion for the previous statement, it is absolutely necessary to carry out the experiment suggested previously, where, at least in the in vitro experiments, the addition of the respective neutralizing antibodies can highlight an effect on gene induction of irradiated cells.

  1. MSC silenced for Hgf or Tsg-6 are just described in the M&M section. Please show the details of their characterization.

 …..Find details of characterization of MSC silenced for Hgf or Tsg-6 48 hours after transfection:

  Gene silencing using siRNA-mediated RNA interference has become a powerful tool in cell biology, addressing a range of research questions. After delivery of siRNA into the cell, rapid degradation of the target mRNA can be detected, often within 24 hours, and subsequently the corresponding protein is also knocked down. The authors measured the strong reduction of specific RNA 48h after transfection, and the results are quite promising.  Nevertheless, the knockdown is transient and, especially in the in vivo experiments, the effects are seen 4-5 weeks after the injection of siRNA-silenced MSC. Is there any way to assess the presence of silenced cells at that timing?

  1. In Fig. 1 the control of MSC supernatant vs non-irradiated fibroblasts is missing.

....For this reason, if we introduced non irradiated cells, we need to introduce supplementary figures without significative results.

From a scientific point of view, the experiment I suggested would exclude that the modifications shown in fig. 3B and 3C are related to a basal effect of MSC rather than a specific effect on the radiation-induced fibrosis. I don’t believe that this is an experiment without significative result.

MINOR

  1. In the legend of Fig. 1 panel A, I suggest inserting the delta symbol between MSC and the deleted gene. The same in Fig. 6 and 7. In panel C a Col3a2 (?) gene is mentioned.

Figure 1A (now 2 A) is the methodological scheme, …….

The delta is just referred to the absence of the specific gene. The use of hMSCs HGF could be interpreted as hMSCs overexpressing HGF, while the symbol D (MSCDHGF/TSG-6) would clarify the absence of HGF. The same result could be obtained by adding si (MSCsiHGF/TSG-6)

  1. TYPOS

In Figure 1 Squared E and squared F are missing

Author Response

Reviewer 1

I believe the authors did a great job of reworking the manuscript. Nevertheless, there is still some questionable points.

  1. The involvement of both HGF and TSG-6 should be evaluated in vitro using neutralizing antibodies.

I cannot scientifically accept the statement of the involvement of TSG-6 and HGF in the inhibition of radiation-induced colonic fibrosis based on the only experiment conducted with silenced MSCs, especially for the prominence that these two molecules have been given, also found in the title.

In my opinion for the previous statement, it is absolutely necessary to carry out the experiment suggested previously, where, at least in the in vitro experiments, the addition of the respective neutralizing antibodies can highlight an effect on gene induction of irradiated cells.

Answer: Silencing of HGF and TSG-6 genes has been used for many years to demonstrate the role of these genes without the use of blocking antibodies.

Compared with conventional assays, siRNA affords high specificity, resulting in improved efficacy and decreased side effects. This technique has been used for more than 10 years to validate the role of a gene.

We justify this below for HGF and TSG-6 in relation to the latest published articles about HGF and TSG-6.  

  1. Of the 13 articles studying the role of TSG-6, the siRNA is used only. Only one study uses siRNA and blocking antibodies, this study concluded that the results are identical.
  2.  
  3. The observation is the same for HGF. See publications bellow.

For TSG-6

  1. Zhang et al Stem Cell Res Ther. 2021; 12: 50. doi: 10.1186/s13287-020-02118-3
  2. Wang S. BMB Rep. 2020 Aug 31; 53(8): 425–430. doi: 10.5483/BMBRep.2020.53.8.268
  3. Reeves SR et al. Front Immunol. 2019; 10: 3159. doi: 10.3389/fimmu.2019.03159
  4. Chaubey et al. Stem Cell Research & Therapy (2018) 9:173. 2018) 9:173 doi.org/10.1186/s13287-018-0903-4
  5. Hertsenberg AJ et al. PLoS One. 2017; 12(3): e0171712. doi: 10.1371/journal.pone.0171712
  6. Magaña-Guerrero FS et al., Sci Rep. 2017; 7: 12426. doi: 10.1038/s41598-017-10962-2
  7. Martin J et al. J Biol Chem. 2016 Jun 24; 291(26): 13789–13801. doi: 10.1074/jbc.M115.670521
  8. Xie J et al. Stem Cells. 2015 Feb; 33(2): 468–478. doi: 10.1002/stem.1851
  9. Wang n et al. Stem Cell Res Ther. 2012; 3(6): 51. doi: 10.1186/scrt142
  10. Zhang S et al. J Biol Chem. 2012 Apr 6; 287(15): 12433–12444. doi: 10.1074/jbc.M112.342873
  11. Foxler DE et al., FEBS Lett. 2011 Apr 6; 585(7): 1089–1096. doi: 10.1016/j.febslet.2011.03.013
  12. Lee RH. et al. Cell Stem Cell. 2009 Jul 2; 5(1): 54–63. doi: 10.1016/j.stem.2009.05.003
  13. Pruitt K et al PLoS Genet. 2006 Mar; 2(3): e40. doi: 10.1371/journal.pgen.0020040
  14.  

For HGF

  1. Zhai Y et Al, Cancer Manag Res. 2020; 12: 11823–11832. doi: 10.2147/CMAR.S277130
  2. Lee HJ et al. JCI Insight. 2020 Jun 18; 5(12): e136059. doi: 10.1172/jci.insight.136059
  3. Yang MH et al. Int J Mol Sci. 2020 Nov; 21(21): 8303. doi: 10.3390/ijms21218303
  4. Nan L et al, Cell Transplant. 2019 Sep; 28(9-10): 1289–1298. doi: 10.1177/0963689719851772
  5. Liu R et al, Oncol Lett. 2018 Nov; 16(5): 5983–5991. doi: 10.3892/ol.2018.9414
  6. Wei W et al. Int J Clin Exp Pathol. 2018; 11(7): 3310–3317. PMID: 31949706
  7. Zhang H. et al, Cancer Sci. 2018 Mar; 109(3): 629–641.   doi: 10.1111/cas.13488
  8. Dilwali S et al, Cancer Biol Ther. 2015 Jan; 16(1): 170–175. doi: 10.4161/15384047.2014.972765
  9. Li XJ et al. Int J Ophthalmol. 2014; 7(2): 239–244. doi: 10.3980/j.issn.2222-3959.2014.02.09
  10. Tsou HK et al. PLoS One. 2013; 8(1): e53974. doi: 10.1371/journal.pone.0053974

Corrections :

  • Lines to 213 to 215: sentence was changed for “(B) Silencing of TSG-6 in MSCs may counteract effect on α-SMA, collagen III and fibronectin expression in irradiated hIFs. (C) Silencing HGF in MSCs may counteract effect on collagen III and fibronectin expression in irradiated hCoSMCs.”
  • Line 302: sentence was changed for “2.6. HGF silencing in transplanted-MSCs may promote fibrosis through a TGF-β-dependent mechanism”
  • Line 402: The consequence was in favor of an increased fibrosis
  • Line 414: Impact of HGF and TSG-6 silencing may highlight importance of secretion of a wide range of factors by therapeutic MSCs.

  1. Gene silencing using siRNA-mediated RNA interference has become a powerful tool in cell biology, addressing a range of research questions. After delivery of siRNA into the cell, rapid degradation of the target mRNA can be detected, often within 24 hours, and subsequently the corresponding protein is also knocked down. The authors measured the strong reduction of specific RNA 48h after transfection, and the results are quite promising. Nevertheless, the knockdown is transient and, especially in the in vivo experiments, the effects are seen 4-5 weeks after the injection of siRNA-silenced MSC. Is there any way to assess the presence of silenced cells at that timing?

Answer: Gene silencing using siRNA-mediated RNA interference has become more and more efficient and stable over time, allowing gene silencing over longer and longer periods. The silencing of our genes is effective over a period of one week, which corresponds to the time of presence of MSCs in the tissue. The rats were injected with MSCs every week, so the effect of the MSCs was renewed every week, as was the silencing of the TSG-6 and HGF genes. It should be noted that we have (and other) published several times that although the injected MSCs disappear after a week, the effect is observable several months after injection.

For exemple in Francois S et al. Stem Cells Transl Med. 2019 Mar;8(3):285-300. doi: 10.1002/sctm.18-0117.

  1. “presence of scarce MSC-GFP-labeled cells (indicated in red) localized in (a) the lamina propria, (b) around the bottom of crypts in the muscularis mucosae and (c) in the sub mucosa, one week after the second MSC injection, but not at later time”
  2. “Interestingly, the presence of exogenous MSCs was observed only at early time points after injection. Therefore, the durable antitumor effects of the MSCs observed is most likely explained by “reprogramming” of resident immune cells by the MSCs, by epigenetic mechanisms, in agreement with the “hit and run” mechanism”

Corrections:

  • Line 457, the following sentence was added “The silencing of HGF and TSG-6 genes is effective over a period of one week.”
  • Line 341: “2.7. TSG-6 may inhibit fibrosis by deregulating macrophage polarization”
  • Line 384: TSG-6 silencing in MSCs may induce an upregulation of myofibroblasts marker α-SMA and genes encoding collagen III and fibronectin. To settle these results, it would be necessary to validate the involvement of HGF and TSG-6 by a neutralizing antibody study.
  • Line 387: “In agreement with previous studies on fibrosis, we observed that HGF and TSG-6 may exert complementary function necessary for anti-fibrotic effects of MSCs during radiation-induced fibrosis (20-22).”

  1. From a scientific point of view, the experiment I suggested would exclude that the modifications shown in fig. 3B and 3C are related to a basal effect of MSC rather than a specific effect on the radiation-induced fibrosis. I don’t believe that this is an experiment without significative result.

Answer: There are many scientific arguments but to keep it short we will present only two that attest to the validity of our results:

  1. It has been known and published for more than 10 years that in the absence of environmental stress, the response of MSCs is weak or insignificant (Chang, P.; et al. Oncotarget 2017, 8, 87821-87836, doi:10.18632/oncotarget.21236.).
  2.  
  3. In our case, the absence of irradiation, profibrosing factors such as SMA or collagens are poorly expressed, as illustrated by the work of Linard et al. below. One can see in C (control) the level of expression of collagens and a-SMA. After irradiation (irr) the expression levels of these genes are strongly increased and it is under these conditions that MSCs can have a repression of these genes. The same is true in our experiments (Linard et al. Stem Cells Transl Med 2018 Aug;7(8):569-582. DOI: 10.1002/sctm.17-0267).
  4.  
  5. Below results from Linard et al.

Title of figure: Bone marrow‐derived mesenchymal stromal cells controlled the fibroblast/myofibroblast accumulation. Real‐time expression of wound healing‐related factors, α‐SMA, TNC, and FN1 (fibronectin).

  • Treatment condition group (non irradiated (C),
  • Irradiated (Irr), suture,
  • Flap without
  • Flap with MSC treatment were sampled close to the scar at the time of surgery for the irradiated

Linard et al.

Corrections:

Lines 193 to 194: “This experiment are in favor of a direct effect of MSCs on irradiated hCoSMCs and irradiated hIFs.”……….” When co-cultured with MSCs, irradiated hIFs revealed a marked reduction in expression of” …………” HGF silencing in MSCs lead to an overexpression of”

Lines 204 to 205: “). Results are in favor of the anti-fibrotic potential of MSCs. We confirmed that both HGF and TSG-6 may play a role in MSC-mediated inhibition of colorectal-fibrosis in vitro.

Minor revision

  1. The delta is just referred to the absence of the specific gene. The use of hMSCs HGF could be interpreted as hMSCs overexpressing HGF, while the symbol D (MSCDHGF/TSG-6) would clarify the absence of HGF. The same result could be obtained by adding si (MSCsiHGF/TSG-6).

Answer: According to recommendation of reviewer 1, we retained to use MSCs siHGF and MSCs siTSG-6

Line 199:   (irradiated + hMSCs siHGF) or silenced for TSG-6

Line 200: (irradiated + hMSCs siTSG-6)

Line 308: Irradiated + MSCs siHGF group

Line 298: Irradiated + MSCs siHGF group group

Line 302: (Irradiated + MSCs siHGF)

Line 313: (Irradiated + MSCs siHGF,

Line 318: siHGF group.

Line 321: Irradiated + MSCs siHGF group

Line 322: Irradiated + MSCs siHGF group

Line 481: (“Irradiated + MSCs siHGF” group) or TSG-6 (“Irradiated + MSCs siTSG-6” group).

Line 514: Rats that received MSC-siHGF or MSC-siTSG-6 (“Irradiated + MSCs siHGF” group and “Irradiated + MSCs siTSG-6” group)

Line 528: MSCs siHGF and Irradiated + MSCs siTSG-6 groups

  1. In the legend of Fig. 1 panel A, I suggest inserting the delta symbol between MSC and the deleted gene. The same in Fig. 6 and 7. In panel C a Col3a2 (?) gene is mentioned.

Answer: Figure 3 (previously Figure 1) and Figure 6 were modified according to reviewer’s suggestions

  1. In Figure 1 Squared E and squared F are missing

Answer: Into Figure 1, squared E and F was added, thank you

Reviewer 2 Report

ijms-1059739

The authors properly response the comments of the reviewers’ and the revised manuscript has been improved. There are a few concerns.

  1. The estimated number of radiation-treated cancer patients still remains obscure. Due to rapid advances of molecular target therapy, including PD-L1 and NTRK, radiation therapy possibly gets decreased even in advanced cancer patients. Please investigate the latest statistics or epidemiological data.
  2. The abbreviation “IBD” possibly stand for “inflammatory bowel disease”, not “intestinal bowel disease”. Please check it.
  3. In the revised manuscript, funding information has been updated. Please add the further detail for IRSN, including the recipient’s name and the grant number if any.

Author Response

Reviewer 2

The authors properly response the comments of the reviewers’ and the revised manuscript has been improved. There are a few concerns.

  1. The estimated number of radiation-treated cancer patients still remains obscure. Due to rapid advances of molecular target therapy, including PD-L1 and NTRK, radiation therapy possibly gets decreased even in advanced cancer patients. Please investigate the latest statistics or epidemiological data.

Line 54, the latest statistical incidence was indicated: The number of new cancer cases worldwide in 2020 was 19 292 789 cases with 9 958 133 deaths (Source: Globocan 2020).”

Line 55, a sentence was added “

“Targeted cancer therapies may be more therapeutically beneficial for lung cancer, colorectal cancer, breast cancer, lymphoma and leukemia. Targeted molecular therapy, such as against Programmed Cell Death Ligand-1 (PD-L1) and genes from the neurotrophin tyrosine kinase receptor (NTRK) family, will probably reduce the number of radiotherapy and will improve the consequence for cancer patients (8)."

  1. The abbreviation “IBD” possibly stand for “inflammatory bowel disease”, not “intestinal bowel disease”. Please check it.

Thank you, sentence was corrected line 616 “inflammatory bowel disease”

  1. In the revised manuscript, funding information has been updated. Please add the further detail for IRSN, including the recipient’s name and the grant number if any.

We apologize for this mistake there is no grant for this research, line 602 we removed funding

Line 642, reference 8: Breen WG, Leventakos K, Dong H, Merrell KW. Radiation and immunotherapy: emerging mechanisms of synergy. Journal of thoracic disease. 2020; 12(11):7011-23. Epub 2020/12/08.

Line 646, Reference 7 was changed by: The Global Cancer Observatory: International Agency for Research on Cancer: World Health Organization 2020 (Globocan). Number of new cases in 2020 both sexes, all ages and number of deaths in 2020, both sexes, all ages. November 2020 (https://gco.iarc.fr/today/data/factsheets/populations/900-world-fact-sheets.pdf).

Round 3

Reviewer 1 Report

Accept in present form